# TimeFK: Towards Time Series Forecasting via Treating LLMs as Fuzzy Key

## Abstract

Time series forecasting (TSF) aims to predict future values based on historical data. Recent advancements in large language models (LLMs), which integrate cross-modal information (time series data and textual prompts), have demonstrated remarkable performance in TSF tasks. However, significant gaps remain between LLM-based methods and deep learning approaches due to their inherent differences. To bridge this gap, we propose TimeFK, an innovative TSF framework that uses LLMs as "fuzzy keys" to activate forecasting capabilities. Specifically, we introduce a tri-branch multi-modal encoding scheme that combines numerical and linguistic representations: (1) a time series encoder generates precise but weak embeddings, (2) a statistical encoder captures robust yet entangled features, and (3) a background encoder learns dataset-related information that remains disentangled. The fusion of precise, robust, and disentangled representations improves prediction accuracy. To further mitigate noise from language prompts, we introduce a Gaussian fuzzy mapping mechanism that maps hidden representations from LLMs into a fuzzy set space, preserving semantic richness while reducing irrelevant noise. Additionally, we prevent entanglement by using fused cross-modal representations as keys and time series embeddings as values in a fuzzy-aware attention decoder, enabling query-based interactions for forecasting. Extensive experiments on seven real-world benchmark datasets demonstrate that TimeFK outperforms state-of-the-art methods, highlighting the effectiveness of integrating fuzzy reasoning with multi-modal time series analysis.

## 1 Introduction

Time series data is ubiquitous across various domains, and accurate forecasting is crucial for decision-making in industries such as finance (Bhambu et al., 2024; Dong et al., 2024), transportation (Kieu et al., 2024; Qin et al., 2024), agriculture (Lv et al., 2024; Liu et al., 2024a), health care (Shen et al., 2024; Chen & Tang, 2025), and climate science (Ma et al., 2023; Valipour et al., 2024). In practice, human experts often rely on multi-modal information to forecast time series. For instance, economists may combine historical market data with policy documents to predict economic trends.

Over the past few decades, time series forecasting (TSF) models have evolved into deep learning approaches, including recurrent neural networks (Hochreiter & Schmidhuber, 1997), convolutional neural networks (Wu et al., 2023), and Transformers (Liu et al., 2024b). These models rely on the design of effective temporal or frequency domain extractors, which achieves high accuracy at the cost of complexity and limited scalability (Yi et al., 2023; Liu et al., 2024b). Recently, the emergence of large language models (LLMs) such as GPT-4 (Sanderson, 2023; Achiam et al., 2023), LLaMA (Touvron et al., 2023; Singh, 2025), and Deepseek-R1 (Guo et al., 2025) have revolutionized NLP tasks and shown promising potential in structured and complex domains (Yan et al., 2024; Jiang et al., 2025). This raises the question: Can LLMs be leveraged for TSF?

As summarized in Table 1, LLMs offer TSF methods a scalable alternative but often face challenges, including insufficient training data, poor numerical reasoning, and loss of linguistic capacity after fine-tuning (Bi et al., 2023; Ma et al., 2024). Recent approaches have explored transforming time series data into textual prompts to better prompt with (or retrieve from) LLMs. However, prompt-based methods tend to introduce textual noise, while retrieval-based approaches encounter design and efficiency bottlenecks (Sun et al., 2024; Liu et al., 2025). Furthermore, tokenizing numerical

Table 1: Comparison for TSF models across different paradigms.

| Category | Method Type | Advantages | Limitations | Representative Works |
|---|---|---|---|---|
| **DL for TSF** | Redesign&Retraining | High accuracy and feature expressiveness | High implementation complexity | iTransformer (Liu et al., 2024b), FreTS (Yi et al., 2023) |
| **LLMs for TSF** | Retraining | Architecture-agnostic and easy to implement | Limited scalability due to insufficient training data | Pretraining (Ma et al., 2024), Earth-Transformer (Bi et al., 2023) |
| | Fine-tuning | Retains prior knowledge with adaptability | Potential degradation of linguistic capabilities | GPT4TS (Zhou et al., 2023), LLM4TS (Chang et al., 2023) |
| **TSF for LLMs** | Retrieval-based | Clear embedding structures | Bottlenecks in inference speed | TimeCMA (Liu et al., 2025), TEST (Sun et al., 2024) |
| | Prompt-based | Enhanced expressiveness via prompts | Entangled representations, prone to prompt noise | TimeLLM (Jin et al., 2024), S2IPLLM (Pan et al., 2024) |
| | | Reduces prompt noise and enhances robustness | Design complexity in fuzzy strategies | **Ours** |

sequences can disrupt numerical precision, reflecting the cognitive distinction between numerical and linguistic reasoning in the human brain (Singh & Strouse, 2024). This leads to a key question: How can we bridge the inherent gap between natural language and numerical time series data to unlock the full potential of LLMs for TSF?

In this work, we aim to harness the prior knowledge of LLMs without modifying their internal parameters, focusing instead on enhancing their utility as TSF augmenters. To this end, we propose TimeFK, a novel framework that leverages fuzzy multi-modal fusion to overcome the challenges of prompt noise and modality fusion. Our method introduces innovative a Gaussian fuzzy mapping mechanism and a fuzzy-aware attention decoder to overcome the limitations of existing approaches, demonstrating the necessity and advantages of our design.

Our main contributions are summarized as follows:

1. We propose a novel tri-branch multi-modal encoder that learns precise, robust, and disentangled time series representations. By combining (i) numerical time series features, (ii) statistical representations, and (iii) background contextual information, the encoder creates a rich cross-modality representation space. We integrate this encoder into the TimeFK framework, enabling LLMs to learn such representations in a unified manner.

2. To reduce noise from language prompts, we introduce a Gaussian fuzzy mapping that projects LLM hidden representations into a fuzzy set space, preserving semantic richness while suppressing irrelevant features.

3. To prevent entangled representations caused by prompt interference, we propose a fuzzy-aware attention decoder. In this decoder, fused cross-modal representations are treated as attention keys and time series embeddings as values within the attention structure, facilitating precise, query-driven downstream predictions.

4. We conduct extensive experiments across seven real-world datasets, showing that TimeFK consistently outperforms state-of-the-art methods on multiple evaluation metrics.

## 2 RELATED WORK

As shown in Table 1, TSF methods can be broadly categorized into three paradigms: traditional deep learning approaches (DL for TSF), LLMs tailored for time series tasks (LLMs for TSF), and time series-enhanced LLMs (TSF for LLMs).

Before the advent of LLMs, deep learning was the dominant TSF approach. Various architectures were proposed to capture complex temporal patterns. NSformer (Liu et al., 2022) improved the modeling of non-stationary time series by introducing stationarization and de-stationarization modules, with both theoretical and empirical validation supporting its effectiveness. FEDformer (Zhou et al.,

2022) incorporated a mixture-of-experts design to enhance trend and seasonal decomposition and proposed a sparse attention mechanism in the frequency domain to balance efficiency and accuracy. TimesNet (Wu et al., 2023) decomposed time series into periodic segments and modeled intra- and inter-period interactions by Inception blocks, facilitating generalized time series modeling. Crossformer (Zhang & Yan, 2023) captured dependencies across both temporal and variable dimensions using an attention mechanism. PatchTST (Nie et al., 2023) patched time series into subsequences, preserving local semantics and enabling long-range temporal modeling. DLinear (Zeng et al., 2023) introduced a simple one-layer linear model that achieved high accuracy through direct multistep prediction. TSMixer (Chen et al., 2023) extracted temporal and feature-wise information by stacking multilayer perceptrons (MLPs) across mixed time-feature dimensions. To address the limitations of pointwise mapping and the information bottleneck of MLP-based approaches, FreTS (Yi et al., 2023) applied MLPs in the frequency domain to improve global dependency modeling. iTransformer (Liu et al., 2024b) leveraged dimension inversion to enhance long-sequence handling, mitigating performance degradation and reducing computational overhead.

Recently, the rise of LLMs has significantly advanced TSF, driven by their large-scale prior parameterization and extensive pretraining on diverse datasets (Bi et al., 2023; Ma et al., 2024; Sun et al., 2024; Liu et al., 2025; Zhou et al., 2023; Chang et al., 2023; Jin et al., 2024; Pan et al., 2024). Research on LLMs for TSF falls into two main subcategories: retraining and fine-tuning. Retraining approaches (Bi et al., 2023; Ma et al., 2024) preserved architectures of LLMs while retraining them from scratch on integrated public/ private TSF datasets, without relying on pretrained checkpoints. The goal was to build time series-specific foundation models capable of emergent reasoning. However, the limited availability of time series data compared to natural language corpora raised concerns about the feasibility of such approaches. In contrast, the fine-tuning approaches (Zhou et al., 2023; Chang et al., 2023) built on the similarity between language modeling and TSF - both involve modeling historical sequences to predict future outcomes, and can be framed as finite-order Markov processes. These approaches adapted pretrained LLMs to TSF by fine-tuning them on time series data. However, fine-tuning may cause LLMs to sacrifice their original linguistic abilities, blurring the line between fine-tuning and retraining.

Instead of modifying the parameters in LLMs, TSF for LLMs approaches preserved linguistic abilities by freezing LLMs and activating their forecasting potential through carefully designed prompts. Prompts have been incorporated as auxiliary inputs to help LLMs interpret TSF and can also be contextualized with time series-related descriptive information or consist of purely textual summaries representing time series patterns (Jin et al., 2024; Pan et al., 2024). However, these approaches often suffered from representation entanglement, where the interleaving of numerical and textual inputs introduces noise and ambiguity. To mitigate this problem, methods such as TimeCMA (Liu et al., 2025) and TEST (Sun et al., 2024) proposed similarity-based retrieval mechanisms to extract clearer time series embeddings from external memory, significantly improving forecasting accuracy. Nonetheless, retrieval-based designs often involved complex architectures and faced inference latency issues, posing challenges for deployment.

## 3 PRELIMINARIES

**Time Series** Let the time series be denoted as $\mathcal{X} \in \mathbb{R}^{L \times D}$, where $L$ represents the length of the lookback window, and $D$ denotes the number of time series variables.

**Prompt Construction** We transform the time series $\mathcal{X}$ into a set of statistical prompts $\mathcal{P} \in \mathbb{R}^{W \times 2D}$, where each prompt $\mathcal{P}_i$ corresponds to a specific variable and consists of $W$ elements that integrate both numerical and linguistic time series values. As illustrated in Figure 1, two prompt sentences are generated for each variable. The first sentence encodes the recent observations of each variable within the lookback window. The second sentence captures the trend, denoted as $\Delta = \sum_{t=1}^{L} \Delta x_t$, which is computed as the cumulative sum of its first-order temporal differences. In addition to prompts derived from the time series, the dataset also includes background information, denoted as $\mathcal{B} \in \mathbb{R}^K$. This serves as contextual metadata that may assist the model in downstream forecasting tasks.

**Problem Definition** Given a lookback window $\mathcal{X} \in \mathbb{R}^{L \times D}$, the objective is to learn a predictive function that utilizes the historical observations $\mathcal{X}$, the corresponding statistical prompts $\mathcal{P}$, and the auxiliary background $\mathcal{B}$ to forecast the future time series $\hat{\mathcal{X}} \in \mathbb{R}^{T \times D}$ over the next $T$ timesteps.

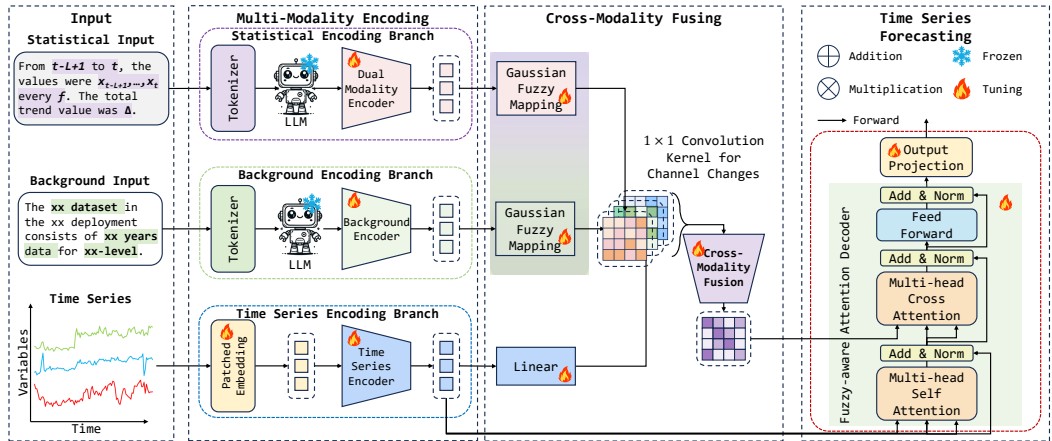

Figure 1: Overview of the TimeFK framework, comprising three modules: Multi-Modality Encoding, Cross-Modality Fusing, and Time Series Forecasting.

## 4 METHOD

To bridge the gap between natural language and time series data, we propose TimeFK, which conceptualizes LLMs as fuzzy keys for TSF (Figure 1). TimeFK consists of three modules: Multi-Modality Encoding, Cross-Modality Fusing, and Time Series Forecasting.

**Multi-Modality Encoding** module consists of three branches: (1) **The time series encoding branch** comprises a patched embedding followed by a time series encoder. The time series of each variable is segmented into patches to generate patch-level token embeddings, which are then passed through a pre-layer normalization Transformer encoder (Xiong et al., 2020). This results in precise yet weak representations of the time series. (2) **The statistical encoding branch** incorporates a frozen pretrained LLM (GPT-2 (Radford et al., 2019) by default) alongside a trainable dual modality encoder. Prompts composed of numerical time series values and their corresponding linguistic trend descriptions are first tokenized and then processed by the LLM, yielding semantically robust yet entangled representations. These representations are further refined by a pre-layer normalization Transformer encoder to enhance dual-modality feature representations. (3) **The background encoding branch** leverages a frozen pretrained LLM and a trainable background encoder to generate disentangled representations that encapsulate semantic priors. These representations are designed to complement the numerical time series data by injecting global contextual knowledge.

Next, we introduce the **Cross-Modality Fusing** module, leveraging a **Gaussian fuzzy mapping** to reduce noise from language prompts while retaining semantic priors. This mechanism maps outputs of the statistical and background encoding branches into a fuzzy set space (Jang, 1993; Zhu et al., 2024), effectively lowering cross-modality interference and improving robustness.

Finally, a **Fuzzy-Aware Attention Decoder** is employed to perform **Time Series Forecasting**. It uses the fused cross-modal representations as keys to guide the decoding, thereby activating its predictive capacity and producing accurate future time series predictions.

### 4.1 MULTI-MODALITY ENCODING

#### 4.1.1 TIME SERIES ENCODING BRANCH

***Patched embedding*** is designed to capture longer-term temporal dependencies, given a multivariate time series $\mathcal{X} \in \mathbb{R}^{L \times D}$, thereby significantly enhancing forecasting performance. First, the input sequence $\mathcal{X}$ is projected into a learnable latent representation $\mathcal{X}_e \in \mathbb{R}^{H \times D}$, facilitating the modeling of complex temporal interactions among variables (Liu et al., 2024b). Next, reversible instance normalization is applied to $\mathcal{X}_e$, transforming it into a normalized representation $\bar{\mathcal{X}}_e$ with zero mean and unit standard deviation. This step effectively mitigates distribution shifts commonly observed

in time series data (Kim et al., 2022). The normalized input is then linearly transformed into the embedding space using trainable parameters $W_e$ and $b_e$, resulting in $\bar{\mathcal{X}}_e = W_e \mathcal{X}_e + b_e$.

After normalization, a patching operation is applied to $\bar{\mathcal{X}}_e$. Each univariate time series is segmented into patches, which may be either overlapping or non-overlapping. Given a patch length of $P$ and a stride of $S$, the patching process produces a sequence of patches, denoted by $\mathcal{X}_p$, where $\mathcal{X}_p \in \mathbb{R}^{P \times C \times D}$. Here, $C = \lfloor \frac{(L-P)}{S} \rfloor + 2$ denotes the total number of patches. To accommodate this segmentation, we pad the input sequence by appending $S$ repeated values of the final timestep.

***Time Series Encoder*** consists of a pre-layer normalization (Pre-LN) Transformer, which offers improved training stability and faster convergence, making it particularly well-suited for TSF (Huang et al., 2023; Liu et al., 2025). In the $i$-th encoder layer, the input patched embedding $\mathcal{X}_p^i$ is first processed by the first layer normalization $\text{LN}_1^i(\cdot)$:

$$\bar{\mathcal{X}}_p^i = \text{LN}_1^i(\mathcal{X}_p^i) = \gamma_1^i \odot \frac{\mathcal{X}_p^i - \mu_1^i}{\sigma_1^i} + \beta_1^i, \tag{1}$$

where $\gamma_1^i$ and $\beta_1^i$ are learnable scaling and shifting parameters, and $\mu_1^i$ and $\sigma_1^i$ denote the mean and standard deviation of $\mathcal{X}_p^i$, respectively. The symbol $\odot$ represents element-wise multiplication.

The normalized representation $\bar{\mathcal{X}}_p^i$ is then passed through a multi-head self-attention mechanism (MHSA) parameterized by $\rho$, followed by a residual connection that adds the original input $\mathcal{X}_p^i$:

$$\mathcal{X}_p^{i+1/2} = \text{MHSA}_\rho^i(\bar{\mathcal{X}}_p^i) + \mathcal{X}_p^i,$$

$$\text{MHSA}_\rho^i(\bar{\mathcal{X}}_p^i) = \rho_o(\text{Attn}_\rho^i(\bar{\mathcal{X}}_p^i)), \ \text{Attn}_\rho^i(\bar{\mathcal{X}}_p^i) = \text{softmax}(\frac{(\rho_q \bar{\mathcal{X}}_p^i)(\rho_k \bar{\mathcal{X}}_p^i)^T}{\sqrt{d_k}})(\rho_v \bar{\mathcal{X}}_p^i), \tag{2}$$

where $\rho_q$, $\rho_k$, $\rho_v$, and $\rho_o$ are linear projection matrices, and $d_k$ denotes the dimension of the vectors after the projection $\rho_k$. The output $\mathcal{X}_p^{i+1/2}$ is subsequently passed through the second layer normalization $\text{LN}_2^i(\cdot)$, parameterized by $\gamma_2^i$ and $\beta_2^i$, followed by a position-wise feed-forward network (FFN) to further refine the representations:

$$\bar{\mathcal{X}}_p^{i+1/2} = \text{LN}_2^i(\mathcal{X}_p^{i+1/2}), \ \mathcal{X}_p^{i+1} = \text{FFN}^i(\bar{\mathcal{X}}_p^{i+1/2}) + \mathcal{X}_p^{i+1/2}. \tag{3}$$

In this work, $\text{FFN}^i(\cdot)$ is implemented as a stack of fully connected layers with ReLU activation (Schmidt-Hieber, 2020). Before cross-modality fusing, the output of the final encoder layer (totally $N_\mathcal{X}$ layers) is projected via a learnable linear transformation to obtain $\widetilde{\mathcal{X}} \in \mathbb{R}^{P \times C \times D}$.

### 4.1.2 STATISTICAL ENCODING BRANCH

***Pretrained LLMs*** are employed to extract semantically robust yet entangled representations. In this study, we adopt GPT-2 as the backbone LLM for generating prompt representations, since decoder-only LLMs such as GPT-2 are significantly more sample-efficient than encoder-only models when trained on the same dataset (BehnamGhader et al., 2024). All parameters of GPT-2 (the tokenizer, positional encoding, and decoder layers) are kept frozen throughout the process.

The tokenizer first maps the statistical prompts $\mathcal{P} \in \mathbb{R}^{W \times 2D}$ into token embeddings $\mathcal{P}_t \in \mathbb{R}^{Q \times D}$, where $Q$ denotes the number of tokens in each prompt. These token embeddings are then passed through the frozen LLM to obtain the contextualized prompt representations $\mathcal{P}_e \in \mathbb{R}^{Q \times D \times H}$, where $H$ is the default hidden dimension of the LLM (768 in GPT-2).

***Dual Modality Encoder*** follows the structure of Pre-LN Transformer encoder, parameterized by $\theta$:

$$\mathcal{P}_e^{j+1/2} = \text{MHSA}_\theta^j(\text{LN}_1^j(\mathcal{P}_e^j)) + \mathcal{P}_e^j, \ \mathcal{P}_e^{j+1} = \text{FFN}^j(\text{LN}_2^j(\mathcal{P}_e^{j+1/2})) + \mathcal{P}_e^{j+1/2}, \tag{4}$$

where $\mathcal{P}_e^{j+1/2}$ represents the hidden states after MHSA and residual connection, while $\mathcal{P}_e^{j+1} \in \mathbb{R}^{Q \times D \times H}$ denotes the output of the $j$-th encoder layer.

Given that not all tokens contribute equally to LLM performance (BehnamGhader et al., 2024; Lin et al., 2024), and considering that the final token in a prompt often captures the most comprehensive linguistic information due to the causal self-attention mechanism of decoder-only LLMs, we retain only the representation of the last token $\widetilde{\mathcal{P}} \in \mathbb{R}^{D \times H}$ from the final ($N_{\mathcal{P}}$-th) layer output. This not only preserves critical semantic information but also reduces computational cost.

### 4.1.3 BACKGROUND ENCODING BRANCH

**Background Encoder** takes as input the background prompt representations $\mathcal{B}_e \in \mathbb{R}^{K \times H}$, which are obtained by mapping the background prompts $\mathcal{B} \in \mathbb{R}^K$ through a frozen LLM, as described in Section 4.1.2. To further extract disentangled semantic representations, $\mathcal{B}_e$ are processed by a Pre-LN Transformer encoder parameterized by $\phi$:

$$\mathcal{B}_e^{m+1/2} = \text{MHSA}_\phi^m(\text{LN}_1^m(\mathcal{B}_e^m)) + \mathcal{B}_e^m, \ \mathcal{B}_e^{m+1} = \text{FFN}^m(\text{LN}_2^m(\mathcal{B}_e^{m+1/2})) + \mathcal{B}_e^{m+1/2}. \quad (5)$$

Unlike the statistical encoding branch, background information serves as a global and abstract summary of the dataset. However, its influence may vary across individual variables. Therefore, to ensure consistent semantic guidance for each variable during the subsequent cross-modality fusing, we enforce the background representations to maintain a shared mean across all variables. To achieve this, the final token output from the last layer (totally $N_{\mathcal{B}}$ layers) of the background encoder is expanded along the $D$-dimensional variable axis, yielding $\widetilde{\mathcal{B}} \in \mathbb{R}^{D \times H}$.

### 4.2 CROSS-MODALITY FUSING

Previous work utilizing frozen LLMs has typically embedded prompts for hidden representation fusion/retrieval, resulting in relatively static representations with limited informativeness under a given prompt. Consequently, prior studies have mainly focused on exploring how prompts can enhance TSF, which introduces noise. To mitigate this noise, we propose to use a **Gaussian fuzzy mapping** mechanism that performs fuzzification on representations generated from the statistical and background encoding branches to estimate the membership values of features. Specifically, the outputs $\widetilde{\mathcal{P}}$ and $\widetilde{\mathcal{B}}$ are passed through a membership function to compute their degrees of association with each patch of the time series:

$$\mu_m(\widetilde{\mathcal{P}}) = f_{\mathcal{P}}(\exp(-\frac{(\widetilde{\mathcal{P}} - c_{\widetilde{\mathcal{P}}})^2}{2\sigma_{\widetilde{\mathcal{P}}}^2})), \ \mu_m(\widetilde{\mathcal{B}}) = f_{\mathcal{B}}(\exp(-\frac{(\widetilde{\mathcal{B}} - c_{\widetilde{\mathcal{B}}})^2}{2\sigma_{\widetilde{\mathcal{B}}}^2})) \quad (6)$$

where $f_{\cdot} : \mathbb{R}^{D \times H} \to \mathbb{R}^{P \times D}$, $\mu_m(\cdot) \in \mathbb{R}^{P \times D}$ denotes the membership values, $c_{\widetilde{\mathcal{P}}}$ and $c_{\widetilde{\mathcal{B}}}$ represent the centers of the Gaussian functions that define the core of each fuzzy set, and $\sigma_{\widetilde{\mathcal{P}}}$, $\sigma_{\widetilde{\mathcal{B}}}$ denote the standard deviations that control the fuzziness of the Gaussian curves. When $\widetilde{\mathcal{P}} = c_{\widetilde{\mathcal{P}}}$, the membership degree reaches its maximum, indicating full inclusion in the set. As $\widetilde{\mathcal{P}}$ deviates from $c_{\widetilde{\mathcal{P}}}$, the membership value decreases toward zero. The same principle applies to $\widetilde{\mathcal{B}}$.

Subsequently, **cross-modality fusion** is applied to integrate the fuzzified statistical and background encodings with the time series features, forming the key input for the subsequent forecasting decoder. This fusion is implemented via a convolutional neural network. Specifically, we concatenate the three sources of information $\mathcal{F}_e = [\mu_m(\widetilde{\mathcal{P}})|\mu_m(\widetilde{\mathcal{B}})|\widetilde{\mathcal{X}}]$, $\mathcal{F}_e \in \mathbb{R}^{P \times (C+2) \times D}$, where $|$ denotes channel-wise concatenation. The final fusion output is computed as $\mathcal{F}_o = \text{CNN}_\eta(\mathcal{F}_e)$, where $\text{CNN}_\eta$ is a $1 \times 1$ convolutional network parameterized by $\eta$, responsible for integrating the time series and multi-modal membership values. The resulting output $\mathcal{F}_o \in \mathbb{R}^{P \times D}$ serves as the fused representations.

### 4.3 TIME SERIES FORECASTING

We design a fuzzy-aware attention decoder for final TSF. The fused representation $\mathcal{F}_o$ serves as the key in a cross-attention mechanism to enhance the decoder's temporal modeling. In the $n$-th layer, the input embedding $\widetilde{\mathcal{X}}^n$ first undergoes self-attention via a multi-head attention (MHSA)

module, followed by residual connection and layer normalization, producing $\widetilde{\mathcal{X}}^{n+1/2}$. It is then passed through multi-head cross-attention (MHCA) with $\mathcal{F}_o$ as key, yielding $\bar{\mathcal{X}}^{n+1/2}$:

$$
\begin{aligned}
\bar{\mathcal{X}}^{n+1/2} &= \text{LN}^n(\text{MHCA}_\xi^n(\widetilde{\mathcal{X}}^{n+1/2}, \mathcal{F}_o) + \widetilde{\mathcal{X}}^{n+1/2}) \\
\text{MHCA}_\xi^n(\widetilde{\mathcal{X}}^{n+1/2}, \mathcal{F}_o) &= \xi_o(\text{Attn}_\xi^n(\widetilde{\mathcal{X}}^{n+1/2}, \mathcal{F}_o)) \\
\text{Attn}_\xi^n(\widetilde{\mathcal{X}}^{n+1/2}, \mathcal{F}_o) &= \text{softmax}(\frac{(\xi_q\widetilde{\mathcal{X}}^{n+1/2})(\xi_k\mathcal{F}_o)^T}{\sqrt{d_k}})(\xi_v\widetilde{\mathcal{X}}^{n+1/2})
\end{aligned}
\tag{7}
$$

where $\xi_q$, $\xi_k$, $\xi_v$, and $\xi_o$ are learnable linear projection matrices for query, key, value, and output transformations, respectively. After MHCA, the representation is further refined via FFN to obtain the final output $\widetilde{\mathcal{X}}^{n+1}$ for the $n$-th decoder layer. The decoder output is then projected through a linear head to produce the final prediction $\hat{\mathcal{X}} \in \mathbb{R}^{T \times D}$.

### 4.4 OVERALL OBJECTIVE FUNCTION

We adopt the Mean Squared Error (MSE) as the training objective to measure the discrepancy between the predicted values and the ground truth. The overall loss function is defined as $\mathcal{L} = \frac{1}{T \times D} \sum_{t=1}^{T} ||\hat{x}_t - x_t||_2^2$, where $T$ denotes the prediction horizon.

## 5 EXPERIMENTS

We evaluate TimeFK on seven widely-used time series datasets: Exchange (Lai et al., 2018), Weather[1], ILI[2], and four ETT datasets (ETTh1, ETTh2, ETTm1, ETTm2) (Zhou et al., 2021). These benchmarks cover diverse temporal dynamics and application scenarios (Wu et al., 2023).

We compare TimeFK against 14 strong baselines, categorized into five groups: (1) Transformer-based deep learning models: iTransformer (Liu et al., 2024b), PatchTST (Nie et al., 2023), Crossformer (Zhang & Yan, 2023), FEDformer (Zhou et al., 2022), NSformer (Liu et al., 2022) and PAttn (Tan et al., 2024). (2) CNN-based deep learning models: TimesNet (Wu et al., 2023). (3)MLP-based deep learning models: DLinear (Zeng et al., 2023), FreTS (Yi et al., 2023) and TSMixer (Chen et al., 2023). (3) Fine-tuning LLMs for TSF: GPT4TS (Zhou et al., 2023). (4) Prompt-based TSF for LLMs: TimeLLM (Jin et al., 2024) and S2IPLLM (Pan et al., 2024). (5) Retrieval-based TSF for LLMs: TimeCMA (Liu et al., 2025). We use two standard evaluation metrics: MSE and Mean Absolute Error (MAE). To ensure a fair comparison, we set the test batch size to 1 for all models. Each experiment is conducted at least three times on NVIDIA A100 GPUs.

### 5.1 MAIN RESULTS

Table 2 summarizes the performance comparison, demonstrating that **TimeFK consistently outperforms all baseline methods across all datasets**. We highlight the following key observations: (1) **Models with LLMs surpass deep learning methods**. This validates our core motivation to leverage LLMs for TSF, as they exhibit stronger generalization and representation capabilities. (2) **The Gaussian fuzzy mapping mechanism is crucial for cross-modality fusion**. On the four ETT datasets, TimeFK achieves superior results due to the introduction of the fuzzy key, which enhances its ability to capture complex cross-modality dependencies and activate TSF capacities. (3) **Prompt-based LLM models outperform Transformer-based deep learning architectures**. Specifically, TimeFK surpasses the best Transformer baseline, iTransformer, with an average improvement of 8.4% in MSE and 5.2% in MAE, indicating that prompt learning significantly enhances time series embeddings. (4) **Compared to other LLM-based methods**, TimeFK achieves an average gain of 13.4% in MSE and 8.0% in MAE, further demonstrating the effectiveness of Gaussian fuzzy mapping mechanism in unlocking the forecasting potential of LLMs. The detailed experimental results of Table 2 are provided in Table 5 of the Appendix.

---

[1]https://www.bgc-jena.mpg.de/wetter/
[2]https://gis.cdc.gov/grasp/fluview/fluportaldashboard.html

Table 2: Average results for TSF. The lowest MSE and MAE values are highlighted in red, while the second-lowest values are highlighted in blue.

| Methods | TimeFK | | TimeCMA | | iTransformer | | PAttn | | TimeLLM | | S2IPLLM | | FreTS | | GPT4TS | | TSMixer | | DLinear | | PatchTST | | Crossformer | | TimesNet | | FEDformer | | NSformer | |
|---|---|---|---|---|---|---|---|---|---|---|---|---|---|---|---|---|---|---|---|---|---|---|---|---|---|---|---|---|---|---|
| Metric | MSE | MAE | MSE | MAE | MSE | MAE | MSE | MAE | MSE | MAE | MSE | MAE | MSE | MAE | MSE | MAE | MSE | MAE | MSE | MAE | MSE | MAE | MSE | MAE | MSE | MAE | MSE | MAE | MSE | MAE |
| ETTh1 | 0.441 | 0.438 | 0.463 | 0.457 | 0.454 | 0.448 | 0.465 | 0.454 | 0.450 | 0.444 | 0.469 | 0.444 | 0.481 | 0.469 | 0.443 | 0.432 | 0.628 | 0.571 | 0.456 | 0.452 | 0.469 | 0.455 | 0.529 | 0.522 | 0.458 | 0.450 | 0.475 | 0.479 | 0.570 | 0.537 |
| ETTh2 | 0.367 | 0.397 | 0.406 | 0.421 | 0.383 | 0.407 | 0.386 | 0.412 | 0.385 | 0.405 | 0.394 | 0.417 | 0.529 | 0.500 | 0.387 | 0.413 | 1.179 | 0.857 | 0.559 | 0.515 | 0.387 | 0.407 | 0.942 | 0.684 | 0.414 | 0.427 | 0.422 | 0.440 | 0.526 | 0.516 |
| ETTm1 | 0.380 | 0.394 | 0.412 | 0.415 | 0.407 | 0.410 | 0.388 | 0.402 | 0.391 | 0.400 | 0.392 | 0.396 | 0.410 | 0.418 | 0.389 | 0.397 | 0.485 | 0.475 | 0.403 | 0.407 | 0.387 | 0.400 | 0.513 | 0.495 | 0.400 | 0.406 | 0.450 | 0.456 | 0.481 | 0.456 |
| ETTm2 | 0.250 | 0.305 | 0.301 | 0.338 | 0.288 | 0.332 | 0.293 | 0.338 | 0.283 | 0.329 | 0.285 | 0.331 | 0.350 | 0.390 | 0.285 | 0.332 | 0.754 | 0.644 | 0.350 | 0.401 | 0.281 | 0.326 | 0.757 | 0.611 | 0.291 | 0.333 | 0.299 | 0.344 | 0.306 | 0.347 |
| Exchange | 0.344 | 0.397 | 0.532 | 0.481 | 0.389 | 0.421 | 0.385 | 0.415 | 0.389 | 0.421 | 0.360 | 0.404 | 0.376 | 0.429 | 0.374 | 0.408 | 0.727 | 0.643 | 0.330 | 0.399 | 0.387 | 0.416 | 1.014 | 0.765 | 0.454 | 0.466 | 0.529 | 0.506 | 0.441 | 0.444 |
| ILI | 2.140 | 0.918 | 2.608 | 1.030 | 2.251 | 0.953 | 2.304 | 0.955 | 3.558 | 1.317 | - | - | 3.389 | 1.269 | 2.774 | 1.103 | 4.788 | 1.511 | 4.060 | 1.459 | 2.321 | 0.954 | 4.838 | 1.496 | 2.338 | 0.947 | 2.781 | 1.125 | 2.695 | 1.047 |
| Weather | 0.248 | 0.275 | 0.251 | 0.283 | 0.258 | 0.280 | 0.255 | 0.279 | 0.276 | 0.294 | 0.270 | 0.288 | 0.251 | 0.297 | 0.266 | 0.285 | 0.249 | 0.313 | 0.265 | 0.316 | 0.255 | 0.278 | 0.251 | 0.310 | 0.269 | 0.293 | 0.346 | 0.392 | 0.322 | 0.331 |
| 1st Count | 12 | | 0 | | 0 | | 0 | | 0 | | 0 | | 0 | | 0 | | 1 | | 0 | | 1 | | 0 | | 0 | | 0 | | 0 | |

Table 3: Average results evaluating the impact of components and branches. The detailed experimental results are provided in Table 6 of the Appendix.

| Methods | TimeFK | | TimeFK w/o LLMs | | TimeFK w/o Patching | | TimeFK w/o Fuzzy | | TimeFK w/o Statistic | | TimeFK w/o Background | |
|---|---|---|---|---|---|---|---|---|---|---|---|---|
| Metric | MSE | MAE | MSE | MAE | MSE | MAE | MSE | MAE | MSE | MAE | MSE | MAE |
| ETTh1 | 0.439 | 0.432 | 0.452 | 0.444 | 0.446 | 0.442 | 0.458 | 0.448 | 0.439 | 0.665 | 0.439 | 0.665 |
| Exchange | 0.344 | 0.397 | 0.361 | 0.404 | 0.365 | 0.408 | 0.347 | 0.399 | 0.411 | 0.561 | 0.411 | 0.561 |
| ILI | 2.140 | 0.918 | 2.600 | 1.056 | 2.635 | 1.066 | 2.563 | 1.050 | 1.017 | 1.566 | 1.016 | 1.565 |
| Weather | 0.248 | 0.275 | 0.253 | 0.279 | 0.255 | 0.280 | 0.252 | 0.279 | 0.277 | 0.497 | 0.279 | 0.498 |

## 5.2 ABLATION STUDY

**Ablation Study on Key Components:** Ablation studies are conducted to evaluate the individual contributions of the key components (LLMs, the patched embedding, and the Gaussian fuzzy mapping) in TimeFK. As shown in Table 3, removing LLMs ("TimeFK w/o LLMs") consistently results in noticeable performance degradation across all datasets. For instance, on the ILI dataset, the MSE increases from 2.140 to 2.600 and MAE from 0.918 to 1.056, indicating that LLMs are essential for capturing complex and irregular temporal dependencies and significantly enhance the model's representational capacity. Similarly, ablating the patched embedding module ("TimeFK w/o Patching") leads to marked declines in performance, particularly in datasets with high temporal variability. On the ILI dataset, MSE rises to 2.635 and MAE to 1.066, confirming that the patched embedding mechanism effectively preserves local temporal patterns and alleviates sequence length limitations, thereby enhancing modeling accuracy. Lastly, removing the Gaussian fuzzy mapping ("TimeFK w/o Fuzzy") also results in performance drops, albeit to a lesser extent. For example, on ETTh1, the MSE increases from 0.439 to 0.458, and MAE from 0.432 to 0.448. While its impact is relatively modest, the fuzzy mapping contributes to modeling uncertainty and soft temporal boundaries, thus playing a complementary role in improving robustness and generalization. These ablation results collectively demonstrate that all three components, LLMs, patched embedding, and Gaussian fuzzy mapping, are integral to the overall performance of TimeFK, with each module addressing different aspects of temporal representation learning.

**Ablation Study on Branches:** To further assess the roles of key architectural branches in TimeFK, we conduct ablation studies by removing the statistical encoding branch ("TimeFK w/o Statistic") and the background encoding branch ("TimeFK w/o Background"). As shown in Table 3, excluding either branch leads to substantial performance degradation across all datasets, particularly in MAE. Removing the statistical encoding branch causes a sharp MAE increase, e.g., from 0.432 to 0.665 on ETTh1, while the MSE on Exchange rises from 0.344 to 0.411. The Weather dataset is most affected, with MAE nearly doubling from 0.275 to 0.497, indicating this branch is crucial for capturing global trends and distributional shifts, especially in stable or long-range sequences. Similarly, removing the background encoding branch results in almost identical degradation, with mirrored MSE/MAE drops across datasets. This suggests a strong interplay between the two branches: the background branch supports coarse-grained temporal modeling and provides essential contextual cues, particularly in complex patterns like those in the ILI dataset (MAE increasing from 0.918 to 1.565). These results highlight that both branches are essential and complementary, jointly enhancing TimeFK's ability to generalize across diverse temporal structures.

Table 4: Average Results on Four Statistical Prompt Designs. Details are provided in Table 7 of the Appendix.

| Methods | Prompt 1 | | Prompt 2 | | Prompt 3 | | Prompt 4 | |
|---|---|---|---|---|---|---|---|---|
| Metric | MSE | MAE | MSE | MAE | MSE | MAE | MSE | MAE |
| ETTh1 | **0.439** | **0.432** | 0.449 | 0.443 | 0.448 | 0.443 | 0.448 | 0.443 |
| Exchange | **0.344** | **0.397** | 0.379 | 0.416 | 0.378 | 0.415 | 0.376 | 0.414 |
| ILI | **2.140** | **0.918** | 2.537 | 1.038 | 2.537 | 1.038 | 2.537 | 1.038 |
| Weather | **0.248** | **0.275** | 0.254 | 0.279 | 0.253 | 0.279 | 0.288 | 0.302 |

## 5.3 MODEL ANALYSIS

### 5.3.1 EFFECT OF STATISTICAL PROMPTS

To assess the impact of statistical prompts on model performance, we compare four variants (Prompt 1-4), as shown in Figure 3. Prompt 1 consistently outperforms others across all datasets, achieving the lowest MSE and MAE, for instance, 0.439/0.432 on ETTh1 and 0.344/0.397 on Exchange, as shown in Table 4. Its advantage is even more evident on datasets with complex temporal patterns, such as ILI (MSE: 2.140 vs. $\geq$2.537 for others). Prompts 2-4 show similar, slightly inferior performance, e.g., MSEs of 0.379, 0.378, and 0.376 on Exchange, all higher than Prompt 1. This suggests Prompt 1 captures trend-sensitive features more effectively. Since statistical prompts are generated dynamically during inference, the Gaussian fuzzy mapping mechanism exhibits limited effectiveness in reducing noise. To better demonstrate the benefits of the Gaussian fuzzy mapping mechanism, we further conduct an analysis focusing on the background prompts.

### 5.3.2 EFFECT OF GAUSSIAN FUZZY MAPPING MECHANISM

We evaluate the Gaussian fuzzy mapping on four benchmarks (ETTh1, Exchange, ILI, Weather) under four background prompt variants generated by ChatGPT. Figure 2 shows MSE and MAE distributions across horizons, with and without the mechanism. As seen in Figure 2 (left), models with fuzzy mapping consistently yield lower MSEs, especially on complex datasets like Exchange and ILI, with reduced variance indicating improved robustness. For example, on Exchange, it markedly lowers both the mean and spread of MSE. Similar gains appear in MAE (right), most pronounced on ILI, highlighting benefits for irregular medical series. Even on smoother datasets (ETTh1, Weather),

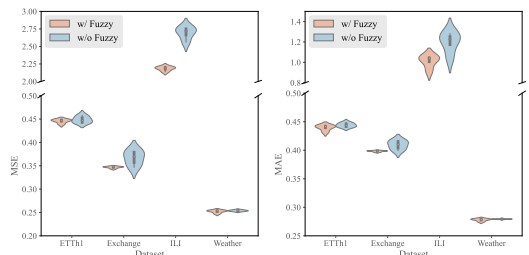

Figure 2: Comparison of w/ and w/o Gaussian fuzzy mapping under four background prompts.

modest error and variance reductions are observed. Overall, Gaussian fuzzy mapping enhances generalization and stability across diverse prompt scenarios, particularly under noisy temporal conditions.

## 6 CONCLUSION

This paper proposes TimeFK, a novel and effective framework for TSF that leverages LLMs as fuzzy keys to activate time series predictive capabilities. Extensive experiments validate the crucial contributions of both the LLMs and the Gaussian fuzzy mapping mechanism in improving forecasting accuracy. Notably, TimeFK outperforms the state-of-the-art transformer-based model, iTransformer, with average improvements of 8.4% in MSE and 5.2% in MAE across multiple benchmark datasets. Furthermore, compared to other LLM-based forecasting approaches, TimeFK achieves an additional reduction of 13.4% in MSE and 8.0% in MAE, demonstrating its superior performance. These findings underscore the potential of integrating fuzzy reasoning with multimodal time series representation, pointing to a promising new direction for future research.

# 7 ETHICS STATEMENT

This work does not involve human subjects, sensitive personal data, or proprietary datasets. All datasets used are publicly available and widely adopted in the time series forecasting literature. We have carefully followed ethical guidelines, including proper citation and licensing terms, and we believe our research does not raise ethical concerns.

# 8 REPRODUCIBILITY STATEMENT

We have made every effort to ensure the reproducibility of our results. Implementation details of the model, training procedures, and hyperparameter settings are provided in Section 4 and Appendix J. All datasets used in our experiments are publicly available, with preprocessing steps consistent with TimesNet (Wu et al., 2023). To further facilitate reproduction, we provide our code and scripts as anonymous supplementary material at `https://anonymous.4open.science/r/TimeFK-B2DF`

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

## A  The Use of Large Language Models (LLMs)

Large Language Models (LLMs) were used solely to assist with language refinement, including grammar correction and typographical error correction. They were not involved in idea generation, data analysis, or interpretation of results. All substantive contributions, including the conception and design of the study, execution of experiments, and manuscript writing, were made exclusively by the authors.

## B  Full Results for Time Series Forecasting

To conserve space in the main text, we report the complete results of the long-term forecasting experiments in Table 5. Notably, S2IPLLM (Pan et al., 2024) produces a NAN prediction on the ILI dataset, and as such, we omit its experimental result from the report.

Table 5: Full results for time series forecasting. We compare extensive competitive models under different prediction lengths. The input sequence length is set to 36 for the ILI dataset and 96 for the others. AVG is averaged from all four prediction lengths. We adopt official codes of baselines and reproduce on our devices for a fair comparison.

| Methods | | TimeFK | | TimeCMA | | iTransformer | | PAttn | | TimeLLM | | S2IPLLM | | FreTS | | GPT4TS | | TSMixer | | DLinear | | PatchTST | | Crossformer | | TimesNet | | FEDformer | | NSformer | |
|---|---|---|---|---|---|---|---|---|---|---|---|---|---|---|---|---|---|---|---|---|---|---|---|---|---|---|---|---|---|---|---|
| Metric | | MSE | MAE | MSE | MAE | MSE | MAE | MSE | MAE | MSE | MAE | MSE | MAE | MSE | MAE | MSE | MAE | MSE | MAE | MSE | MAE | MSE | MAE | MSE | MAE | MSE | MAE | MSE | MAE | MSE | MAE |
| ETTh1 | 96 | 0.379 | 0.401 | 0.402 | 0.420 | 0.386 | 0.405 | 0.398 | 0.411 | 0.388 | 0.403 | 0.397 | 0.403 | 0.400 | 0.412 | 0.379 | 0.394 | 0.497 | 0.489 | 0.386 | 0.400 | 0.414 | 0.419 | 0.423 | 0.448 | 0.384 | 0.402 | 0.392 | 0.429 | 0.513 | 0.491 |
| | 192 | 0.434 | 0.431 | 0.457 | 0.452 | 0.441 | 0.436 | 0.449 | 0.444 | 0.436 | 0.433 | 0.454 | 0.432 | 0.451 | 0.443 | 0.434 | 0.424 | 0.584 | 0.540 | 0.437 | 0.432 | 0.460 | 0.445 | 0.471 | 0.474 | 0.436 | 0.429 | 0.446 | 0.460 | 0.534 | 0.504 |
| | 336 | 0.475 | 0.451 | 0.494 | 0.469 | 0.487 | 0.458 | 0.510 | 0.476 | 0.476 | 0.454 | 0.502 | 0.456 | 0.509 | 0.481 | 0.476 | 0.444 | 0.676 | 0.595 | 0.481 | 0.459 | 0.501 | 0.466 | 0.570 | 0.546 | 0.491 | 0.469 | 0.504 | 0.490 | 0.588 | 0.535 |
| | 720 | 0.477 | 0.471 | 0.502 | 0.487 | 0.503 | 0.491 | 0.502 | 0.486 | 0.499 | 0.486 | 0.523 | 0.486 | 0.563 | 0.540 | 0.482 | 0.467 | 0.754 | 0.659 | 0.519 | 0.516 | 0.500 | 0.488 | 0.653 | 0.621 | 0.521 | 0.500 | 0.558 | 0.536 | 0.643 | 0.616 |
| | AVG | 0.441 | 0.438 | 0.463 | 0.457 | 0.454 | 0.448 | 0.465 | 0.454 | 0.450 | 0.444 | 0.469 | 0.444 | 0.481 | 0.469 | 0.443 | 0.432 | 0.628 | 0.571 | 0.456 | 0.452 | 0.469 | 0.485 | 0.529 | 0.522 | 0.458 | 0.450 | 0.475 | 0.479 | 0.570 | 0.537 |
| ETTh2 | 96 | 0.291 | 0.344 | 0.333 | 0.370 | 0.297 | 0.349 | 0.300 | 0.352 | 0.301 | 0.348 | 0.320 | 0.370 | 0.342 | 0.397 | 0.303 | 0.355 | 0.874 | 0.736 | 0.333 | 0.387 | 0.302 | 0.348 | 0.745 | 0.584 | 0.340 | 0.374 | 0.341 | 0.384 | 0.476 | 0.458 |
| | 192 | 0.366 | 0.391 | 0.422 | 0.422 | 0.380 | 0.400 | 0.376 | 0.399 | 0.381 | 0.395 | 0.395 | 0.411 | 0.440 | 0.449 | 0.385 | 0.404 | 1.192 | 0.881 | 0.477 | 0.476 | 0.388 | 0.400 | 0.877 | 0.656 | 0.402 | 0.414 | 0.413 | 0.427 | 0.512 | 0.493 |
| | 336 | 0.407 | 0.423 | 0.426 | 0.439 | 0.428 | 0.432 | 0.420 | 0.434 | 0.426 | 0.431 | 0.428 | 0.437 | 0.538 | 0.509 | 0.420 | 0.435 | 1.276 | 0.897 | 0.594 | 0.541 | 0.426 | 0.433 | 1.043 | 0.731 | 0.452 | 0.452 | 0.454 | 0.464 | 0.552 | 0.551 |
| | 720 | 0.402 | 0.430 | 0.441 | 0.455 | 0.427 | 0.445 | 0.448 | 0.462 | 0.431 | 0.445 | 0.432 | 0.449 | 0.796 | 0.644 | 0.439 | 0.457 | 1.373 | 0.913 | 0.831 | 0.657 | 0.431 | 0.446 | 1.104 | 0.763 | 0.462 | 0.468 | 0.481 | 0.486 | 0.562 | 0.560 |
| | AVG | 0.367 | 0.397 | 0.406 | 0.421 | 0.383 | 0.407 | 0.386 | 0.412 | 0.385 | 0.405 | 0.394 | 0.417 | 0.529 | 0.500 | 0.387 | 0.413 | 1.179 | 0.857 | 0.559 | 0.515 | 0.387 | 0.407 | 0.942 | 0.684 | 0.414 | 0.427 | 0.422 | 0.440 | 0.526 | 0.516 |
| ETTm1 | 96 | 0.317 | 0.357 | 0.338 | 0.375 | 0.334 | 0.368 | 0.324 | 0.365 | 0.331 | 0.368 | 0.331 | 0.363 | 0.340 | 0.375 | 0.332 | 0.365 | 0.432 | 0.438 | 0.345 | 0.372 | 0.329 | 0.367 | 0.404 | 0.426 | 0.338 | 0.375 | 0.381 | 0.424 | 0.386 | 0.398 |
| | 192 | 0.367 | 0.386 | 0.386 | 0.402 | 0.377 | 0.391 | 0.373 | 0.390 | 0.377 | 0.386 | 0.367 | 0.380 | 0.383 | 0.398 | 0.367 | 0.382 | 0.446 | 0.451 | 0.380 | 0.389 | 0.367 | 0.385 | 0.450 | 0.451 | 0.374 | 0.387 | 0.429 | 0.445 | 0.459 | 0.444 |
| | 336 | 0.406 | 0.411 | 0.424 | 0.422 | 0.426 | 0.420 | 0.395 | 0.408 | 0.401 | 0.407 | 0.402 | 0.402 | 0.420 | 0.425 | 0.397 | 0.403 | 0.491 | 0.481 | 0.413 | 0.413 | 0.399 | 0.410 | 0.532 | 0.515 | 0.410 | 0.411 | 0.463 | 0.463 | 0.495 | 0.464 |
| | 720 | 0.428 | 0.421 | 0.502 | 0.461 | 0.491 | 0.459 | 0.459 | 0.446 | 0.454 | 0.440 | 0.467 | 0.439 | 0.497 | 0.474 | 0.460 | 0.438 | 0.572 | 0.529 | 0.474 | 0.453 | 0.454 | 0.439 | 0.666 | 0.589 | 0.478 | 0.450 | 0.528 | 0.494 | 0.585 | 0.516 |
| | AVG | 0.380 | 0.394 | 0.412 | 0.415 | 0.407 | 0.410 | 0.388 | 0.402 | 0.391 | 0.400 | 0.392 | 0.396 | 0.410 | 0.418 | 0.389 | 0.397 | 0.485 | 0.475 | 0.403 | 0.407 | 0.387 | 0.400 | 0.513 | 0.495 | 0.400 | 0.406 | 0.450 | 0.456 | 0.481 | 0.456 |
| ETTm2 | 96 | 0.174 | 0.256 | 0.188 | 0.270 | 0.180 | 0.264 | 0.184 | 0.268 | 0.179 | 0.264 | 0.180 | 0.265 | 0.195 | 0.285 | 0.177 | 0.263 | 0.244 | 0.355 | 0.193 | 0.292 | 0.175 | 0.259 | 0.287 | 0.366 | 0.187 | 0.267 | 0.194 | 0.282 | 0.192 | 0.274 |
| | 192 | 0.178 | 0.260 | 0.261 | 0.316 | 0.250 | 0.309 | 0.248 | 0.308 | 0.243 | 0.304 | 0.247 | 0.307 | 0.277 | 0.351 | 0.243 | 0.306 | 0.385 | 0.471 | 0.284 | 0.362 | 0.241 | 0.302 | 0.414 | 0.492 | 0.249 | 0.309 | 0.264 | 0.322 | 0.280 | 0.339 |
| | 336 | 0.245 | 0.305 | 0.321 | 0.354 | 0.311 | 0.348 | 0.315 | 0.353 | 0.306 | 0.345 | 0.304 | 0.343 | 0.370 | 0.404 | 0.313 | 0.351 | 0.685 | 0.656 | 0.369 | 0.427 | 0.305 | 0.343 | 0.597 | 0.542 | 0.321 | 0.353 | 0.321 | 0.351 | 0.334 | 0.361 |
| | 720 | 0.404 | 0.399 | 0.434 | 0.414 | 0.412 | 0.407 | 0.427 | 0.422 | 0.404 | 0.401 | 0.408 | 0.407 | 0.560 | 0.519 | 0.407 | 0.407 | 1.704 | 1.095 | 0.554 | 0.522 | 0.402 | 0.400 | 1.730 | 1.042 | 0.408 | 0.403 | 0.418 | 0.415 | 0.417 | 0.413 |
| | AVG | 0.250 | 0.305 | 0.301 | 0.338 | 0.288 | 0.332 | 0.293 | 0.338 | 0.283 | 0.329 | 0.285 | 0.331 | 0.350 | 0.390 | 0.285 | 0.329 | 0.754 | 0.644 | 0.350 | 0.401 | 0.281 | 0.326 | 0.797 | 0.611 | 0.291 | 0.333 | 0.299 | 0.334 | 0.306 | 0.347 |
| Exchange | 96 | 0.083 | 0.203 | 0.119 | 0.241 | 0.095 | 0.217 | 0.087 | 0.205 | 0.087 | 0.205 | 0.081 | 0.200 | 0.089 | 0.217 | 0.085 | 0.202 | 0.199 | 0.361 | 0.083 | 0.209 | 0.090 | 0.208 | 0.590 | 0.582 | 0.129 | 0.259 | 0.153 | 0.282 | 0.125 | 0.255 |
| | 192 | 0.176 | 0.297 | 0.220 | 0.340 | 0.183 | 0.306 | 0.185 | 0.306 | 0.187 | 0.306 | 0.180 | 0.302 | 0.251 | 0.367 | 0.172 | 0.294 | 0.579 | 0.601 | 0.180 | 0.301 | 0.180 | 0.303 | 0.546 | 0.562 | 0.235 | 0.353 | 0.280 | 0.386 | 0.234 | 0.346 |
| | 336 | 0.319 | 0.409 | 0.452 | 0.498 | 0.350 | 0.430 | 0.335 | 0.421 | 0.366 | 0.438 | 0.349 | 0.428 | 0.367 | 0.454 | 0.347 | 0.424 | 0.488 | 0.551 | 0.302 | 0.413 | 0.356 | 0.432 | 1.127 | 0.839 | 0.389 | 0.462 | 0.487 | 0.513 | 0.422 | 0.473 |
| | 720 | 0.797 | 0.677 | 1.334 | 0.847 | 0.930 | 0.732 | 0.931 | 0.726 | 0.918 | 0.734 | 0.829 | 0.685 | 0.795 | 0.677 | 0.891 | 0.712 | 1.642 | 1.058 | 0.763 | 0.665 | 0.922 | 0.721 | 1.792 | 1.078 | 1.063 | 0.790 | 1.196 | 0.844 | 0.981 | 0.701 |
| | AVG | 0.344 | 0.397 | 0.532 | 0.481 | 0.389 | 0.421 | 0.385 | 0.415 | 0.389 | 0.421 | 0.360 | 0.404 | 0.376 | 0.429 | 0.374 | 0.408 | 0.727 | 0.643 | 0.330 | 0.399 | 0.387 | 0.416 | 1.014 | 0.765 | 0.454 | 0.466 | 0.529 | 0.506 | 0.441 | 0.444 |
| ILI | 24 | 2.446 | 0.926 | 3.326 | 1.145 | 2.339 | 0.927 | 2.279 | 0.949 | 4.300 | 1.509 | - | - | 3.316 | 1.268 | 3.067 | 1.194 | 4.530 | 1.479 | 4.320 | 1.551 | 2.329 | 0.949 | 4.809 | 1.492 | 2.433 | 0.936 | 2.771 | 1.140 | 2.583 | 1.004 |
| | 36 | 2.032 | 0.915 | 2.749 | 1.041 | 2.186 | 0.964 | 2.118 | 0.931 | 2.389 | 0.957 | - | - | 3.195 | 1.214 | 2.653 | 1.066 | 4.336 | 1.428 | 4.015 | 1.457 | 2.438 | 0.968 | 4.831 | 1.486 | 2.226 | 0.904 | 2.787 | 1.118 | 2.770 | 1.058 |
| | 48 | 2.054 | 0.904 | 2.234 | 0.961 | 2.319 | 0.953 | 3.232 | 1.236 | | | - | - | 3.313 | 1.250 | 2.653 | 1.068 | 4.908 | 1.530 | 3.826 | 1.399 | 2.278 | 0.945 | 4.608 | 1.457 | 2.329 | 0.968 | 2.709 | 1.100 | 2.718 | 1.057 |
| | 60 | 2.026 | 0.925 | 2.169 | 0.971 | 2.315 | 0.994 | 2.230 | 0.964 | 3.235 | 1.235 | - | - | 3.731 | 1.346 | 2.718 | 1.083 | 5.377 | 1.607 | 4.077 | 1.429 | 2.237 | 0.957 | 5.103 | 1.547 | 2.365 | 0.982 | 2.857 | 1.140 | 2.709 | 1.069 |
| | AVG | 2.140 | 0.918 | 2.608 | 1.030 | 2.251 | 0.953 | 2.304 | 0.955 | 3.558 | 1.317 | - | - | 3.389 | 1.269 | 2.774 | 1.103 | 4.788 | 1.511 | 4.060 | 1.459 | 2.321 | 0.954 | 4.838 | 1.496 | 2.338 | 0.947 | 2.781 | 1.125 | 2.695 | 1.047 |
| Weather | 96 | 0.159 | 0.202 | 0.170 | 0.216 | 0.173 | 0.213 | 0.174 | 0.216 | 0.197 | 0.236 | 0.191 | 0.230 | 0.174 | 0.229 | 0.185 | 0.224 | 0.165 | 0.244 | 0.195 | 0.254 | 0.173 | 0.215 | 0.158 | 0.241 | 0.200 | 0.242 | 0.225 | 0.306 | 0.207 | 0.252 |
| | 192 | 0.210 | 0.252 | 0.215 | 0.258 | 0.221 | 0.254 | 0.220 | 0.257 | 0.242 | 0.275 | 0.236 | 0.267 | 0.213 | 0.265 | 0.230 | 0.263 | 0.216 | 0.291 | 0.235 | 0.293 | 0.219 | 0.256 | 0.201 | 0.281 | 0.232 | 0.271 | 0.346 | 0.400 | 0.272 | 0.296 |
| | 336 | 0.270 | 0.296 | 0.276 | 0.306 | 0.281 | 0.299 | 0.276 | 0.297 | 0.298 | 0.312 | 0.290 | 0.304 | 0.266 | 0.312 | 0.285 | 0.301 | 0.277 | 0.344 | 0.281 | 0.331 | 0.274 | 0.295 | 0.266 | 0.325 | 0.283 | 0.304 | 0.403 | 0.437 | 0.363 | 0.359 |
| | 720 | 0.351 | 0.349 | 0.343 | 0.351 | 0.357 | 0.351 | 0.351 | 0.346 | 0.366 | 0.353 | 0.352 | 0.349 | 0.349 | 0.381 | 0.361 | 0.376 | 0.347 | 0.385 | 0.338 | 0.376 | 0.352 | 0.346 | 0.378 | 0.394 | 0.359 | 0.354 | 0.412 | 0.424 | 0.418 | 0.418 |
| | AVG | 0.248 | 0.275 | 0.251 | 0.283 | 0.258 | 0.280 | 0.255 | 0.279 | 0.276 | 0.294 | 0.270 | 0.288 | 0.251 | 0.297 | 0.266 | 0.285 | 0.249 | 0.313 | 0.265 | 0.316 | 0.255 | 0.278 | 0.251 | 0.310 | 0.269 | 0.293 | 0.346 | 0.392 | 0.322 | 0.331 |
| 1st Count | | 44 | | 0 | | 0 | | 2 | | 0 | | 5 | | 0 | | 9 | | 1 | | 5 | | 3 | | 3 | | 1 | | 0 | | 0 | |

## C  Full Results of the Ablation Study on LLMs, Patched Embedding and Gaussian Fuzzy Mapping

The detailed ablation study results, assessing the effects of LLMs, patched embedding, and Gaussian fuzzy mapping, are shown in Table 6, as they could not be included in the main text due to space constraints.

## D  Full Results on Four Statistical Prompt Designs

To further investigate the influence of statistical prompt engineering on model performance, we conduct a comparative analysis of four statistical prompt variants, denoted as Prompt 1 through Prompt 4, as presented in Figure 3 and Table 7.

Among all variants, Prompt 1 consistently achieves the best performance across all datasets, yielding the lowest MSE and MAE values. For example, on the ETTh1 dataset, Prompt 1 achieves an MSE of 0.439 and an MAE of 0.432, the best results among all prompt types. A similar trend is observed on the Exchange dataset, where Prompt 1 attains an MSE of 0.344 and an MAE of 0.397. The advantage becomes even more

```
From t-L+1 to t, the values
were x_{t-L+1},...,x_t every f. The
total trend value was Δ.
```
Prompt 1: frequency highlighted
```
From t-L+1 to t, the values
were x_{t-L+1},...,x_t every f.
```
Prompt 2: only time series
```
From t-L+1 to t, the values
were x_{t-L+1},...,x_t every f. The
average value was μ.
```
Prompt 3: calculate average
```
From t-L+1 to t, the values
were x_{t-L+1},...,x_t every f. The
total number of historical
hours was T.
```
Prompt 4: given history length

Figure 3: Four Statistical Prompt Designs.

Table 6: Full results evaluating the impact of components and branches.

| Methods | | TimeFK | | TimeFK w/o LLMs | | TimeFK w/o Patching | | TimeFK w/o Fuzzy | | TimeFK w/o Statistic | | TimeFK w/o Background | |
|---|---|---|---|---|---|---|---|---|---|---|---|---|---|
| Metric | | MSE | MAE | MSE | MAE | MSE | MAE | MSE | MAE | MSE | MAE | MSE | MAE |
| ETTh1 | 96 | **0.379** | **0.394** | 0.385 | 0.402 | 0.385 | 0.403 | 0.389 | 0.405 | 0.401 | 0.618 | 0.401 | 0.618 |
| | 192 | **0.430** | **0.424** | 0.434 | 0.431 | 0.438 | 0.434 | 0.443 | 0.436 | 0.432 | 0.659 | 0.432 | 0.659 |
| | 336 | 0.469 | **0.444** | 0.477 | 0.453 | 0.478 | 0.455 | 0.483 | 0.458 | **0.450** | 0.688 | **0.450** | 0.688 |
| | 720 | 0.479 | **0.467** | 0.514 | 0.489 | 0.482 | 0.475 | 0.516 | 0.494 | 0.475 | 0.696 | **0.474** | 0.695 |
| | AVG | **0.439** | **0.432** | 0.452 | 0.444 | 0.446 | 0.442 | 0.458 | 0.448 | **0.439** | 0.665 | **0.439** | 0.665 |
| Exchange | 96 | **0.083** | **0.203** | 0.086 | 0.205 | 0.085 | 0.205 | 0.084 | 0.205 | 0.208 | 0.297 | 0.207 | 0.296 |
| | 192 | **0.176** | **0.297** | 0.177 | 0.300 | 0.183 | 0.306 | 0.177 | 0.304 | 0.307 | 0.431 | 0.308 | 0.434 |
| | 336 | **0.319** | **0.409** | 0.332 | 0.417 | 0.358 | 0.432 | 0.321 | 0.411 | 0.409 | 0.567 | 0.409 | 0.567 |
| | 720 | 0.797 | **0.677** | 0.848 | 0.696 | 0.834 | 0.688 | 0.805 | 0.678 | **0.720** | 0.948 | 0.721 | 0.949 |
| | AVG | **0.344** | **0.397** | 0.361 | 0.404 | 0.365 | 0.408 | 0.347 | 0.399 | 0.411 | 0.561 | 0.411 | 0.561 |
| ILI | 24 | **2.446** | **0.926** | 2.777 | 1.094 | 2.710 | 1.075 | 2.615 | 1.057 | 1.051 | 1.625 | 1.052 | 1.625 |
| | 36 | **2.032** | **0.915** | 2.471 | 1.022 | 2.530 | 1.038 | 2.486 | 1.031 | 0.994 | 1.534 | 0.994 | 1.533 |
| | 48 | **2.054** | **0.904** | 2.534 | 1.039 | 2.564 | 1.048 | 2.505 | 1.034 | 0.989 | 1.525 | 0.988 | 1.523 |
| | 60 | 2.026 | **0.925** | 2.618 | 1.067 | 2.736 | 1.103 | 2.646 | 1.079 | **1.033** | 1.579 | 1.032 | 1.578 |
| | AVG | 2.140 | **0.918** | 2.600 | 1.056 | 2.635 | 1.066 | 2.563 | 1.050 | 1.017 | 1.566 | **1.016** | 1.565 |
| Weather | 96 | **0.159** | **0.202** | 0.171 | 0.214 | 0.172 | 0.213 | 0.166 | 0.210 | 0.209 | 0.407 | 0.210 | 0.408 |
| | 192 | **0.210** | **0.252** | 0.218 | 0.256 | 0.220 | 0.257 | 0.217 | 0.257 | 0.256 | 0.467 | 0.257 | 0.468 |
| | 336 | **0.270** | **0.296** | 0.273 | 0.298 | 0.276 | 0.298 | 0.275 | 0.300 | 0.297 | 0.522 | 0.299 | 0.524 |
| | 720 | 0.351 | **0.349** | 0.350 | **0.349** | 0.353 | 0.350 | 0.350 | 0.351 | **0.347** | 0.591 | 0.349 | 0.592 |
| | AVG | **0.248** | **0.275** | 0.253 | 0.279 | 0.255 | 0.280 | 0.252 | 0.279 | 0.277 | 0.497 | 0.279 | 0.498 |

pronounced on datasets with complex temporal dynamics, such as ILI, where Prompt 1 reduces the MSE to 2.140, compared to 2.537 for the other prompts. Prompts 2, 3, and 4 exhibit similar performance trends, with only marginal differences among them. For instance, on the Exchange dataset, the MSE values for Prompts 2, 3, and 4 are 0.379, 0.378, and 0.376, respectively, all clearly higher than that of Prompt 1. This consistent performance gap suggests that, while Prompts 2-4 capture certain temporal dependencies, they may lack the trend-sensitive representations embedded in Prompt 1.

Notably, since statistical prompts are dynamically generated during inference based on the input time series, especially in terms of values and evolving trends, the effectiveness of different prompts varies significantly. As a result, the choice of prompt has a substantial impact on forecasting accuracy, particularly reflected in MSE and MAE.

Table 7: Full Results on four statistical prompt designs

| Methods | | Prompt 1 | | Prompt 2 | | Prompt 3 | | Prompt 4 | |
|---|---|---|---|---|---|---|---|---|---|
| Metric | | MSE | MAE | MSE | MAE | MSE | MAE | MSE | MAE |
| ETTh1 | 96 | **0.379** | **0.394** | 0.382 | 0.402 | 0.382 | 0.401 | 0.383 | 0.402 |
| | 192 | **0.430** | **0.424** | 0.437 | 0.433 | 0.434 | 0.432 | 0.434 | 0.432 |
| | 336 | **0.469** | **0.444** | 0.487 | 0.458 | 0.486 | 0.458 | 0.488 | 0.458 |
| | 720 | **0.479** | **0.467** | 0.488 | 0.480 | 0.488 | 0.480 | 0.486 | 0.478 |
| | AVG | **0.439** | **0.432** | 0.449 | 0.443 | 0.448 | 0.443 | 0.448 | 0.443 |
| Exchange | 96 | **0.083** | **0.203** | 0.088 | 0.207 | 0.085 | 0.204 | 0.088 | 0.207 |
| | 192 | **0.176** | **0.297** | 0.180 | 0.304 | 0.180 | 0.304 | 0.179 | 0.303 |
| | 336 | **0.319** | **0.409** | 0.378 | 0.447 | 0.375 | 0.445 | 0.371 | 0.442 |
| | 720 | **0.797** | **0.677** | 0.868 | 0.705 | 0.871 | 0.706 | 0.868 | 0.704 |
| | AVG | **0.344** | **0.397** | 0.379 | 0.416 | 0.378 | 0.415 | 0.376 | 0.414 |
| ILI | 24 | **2.446** | **0.926** | 2.654 | 1.055 | 2.654 | 1.055 | 2.654 | 1.056 |
| | 36 | **2.032** | **0.915** | 2.434 | 1.013 | 2.434 | 1.013 | 2.434 | 1.013 |
| | 48 | **2.054** | **0.904** | 2.473 | 1.024 | 2.474 | 1.024 | 2.474 | 1.024 |
| | 60 | **2.026** | **0.925** | 2.585 | 1.058 | 2.586 | 1.058 | 2.586 | 1.058 |
| | AVG | **2.140** | **0.918** | 2.537 | 1.038 | 2.537 | 1.038 | 2.537 | 1.038 |
| Weather | 96 | **0.159** | **0.202** | 0.171 | 0.213 | 0.169 | 0.212 | 0.168 | 0.211 |
| | 192 | **0.210** | **0.252** | 0.219 | 0.257 | 0.220 | 0.257 | 0.279 | 0.300 |
| | 336 | **0.270** | **0.296** | 0.273 | 0.298 | 0.274 | 0.299 | 0.354 | 0.349 |
| | 720 | **0.351** | **0.349** | 0.352 | 0.349 | 0.351 | 0.348 | 0.351 | 0.348 |
| | AVG | **0.248** | **0.275** | 0.254 | 0.279 | 0.253 | 0.279 | 0.288 | 0.302 |

# E FULL RESULTS OF IMPACT ON BACKGROUND PROMPTS

We construct four distinct background prompts to investigate the impact of textual context on model performance. All prompts were generated using ChatGPT. Prompt 1 corresponds to the default background prompt used in TimeLLM, which describes each dataset as Figure 4. Prompts 2-4 are semantically equivalent paraphrases generated by LLMs to evaluate model robustness under different linguistic formulations. These variations are designed to maintain the same factual content while diversifying the expression style, as shown in Figure 5-7.

```
ETT: The Electricity Transformer Temperature
(ETT) is a crucial indicator in the electric
power long-term deployment. This dataset
consists of 2 years data from two separated
counties in China. To explore the granularity
on the Long sequence time-series forecasting
(LSTF) problem, different subsets are created,
{ETTh1, ETTh2} for 1-hour-level and ETTm1 for
15-minutes-level. Each data point consists of
the target value "oil temperature" and 6 power
load features. The train/val/test is 12/4/4
months.
Exchange: The collection of the daily exchange
rates of eight foreign countries including
Australia, British, Canada, Switzerland, China,
Japan, New Zealand and Singapore ranging from
1990 to 2016.
ILI: The influenza-like illness (ILI) dataset
records data on patients with ILI recorded
weekly by the Centers for Disease Control and
Prevention from 2002 to 2021, which describes
the ratio of ILI patients to the total number
of patients.
Weather: Weather is recorded every 10 minutes
for the 2020 whole year, which contains 21
meteorological indicators, such as air
temperature, humidity, etc.
```

Figure 4: Background Prompt 1.

```
ETT: The Electricity Transformer Temperature
(ETT) dataset, reflecting key metrics for long-
term electricity management, includes two years
of data from two counties in China. To examine
time-series forecasting at different
granularities, subsets like ETTh1 and ETTh2
(hourly) and ETTm1 (15-minute intervals) are
provided. Each entry includes 'oil temperature'
as the target variable along with six power
load indicators. The dataset is split into
training, validation, and testing sets using a
12/4/4-month ratio.
Exchange: This dataset contains daily foreign
exchange rates from 1990 to 2016 for eight
countries: Australia, the UK, Canada,
Switzerland, China, Japan, New Zealand, and
Singapore.
ILI: The ILI dataset offers weekly records from
2002 to 2021, compiled by the CDC, showing the
proportion of influenza-like illness cases
among the total number of patients.
Weather: This dataset provides weather data
collected at 10-minute intervals throughout the
year 2020, including 21 meteorological
variables such as temperature, humidity, and
more.
```

Figure 5: Background Prompt 2.

**ETT**: The Electricity Transformer Temperature (ETT) dataset is essential for power grid planning. It features two years of hourly data from two Chinese counties. To support long-sequence forecasting, versions like ETTh1 and ETTh2 (hourly), and ETTm1 (15-minute intervals) are offered. Each entry records the oil temperature (target) along with six associated power load variables. Data is partitioned into 12 months for training, 4 months for validation, and 4 months for testing.
**Exchange**: It includes daily currency exchange rates from 1990 to 2016 across eight countries—Australia, the United Kingdom, Canada, Switzerland, China, Japan, New Zealand, and Singapore.
**ILI**: Collected by the CDC from 2002 to 2021, this dataset contains weekly information on influenza-like illness cases, expressed as the proportion of ILI cases among all patient visits.
**Weather**: This dataset captures weather conditions every 10 minutes throughout 2020, encompassing 21 different meteorological attributes such as temperature, humidity, and more.

Figure 6: Background Prompt 3.

**ETT**: The Electricity Transformer Temperature (ETT) dataset is an important benchmark for long-horizon time-series forecasting in energy systems. It comprises two years of hourly resolution data collected from two distinct counties in China. To accommodate various temporal granularities, three subsets are defined: ETTh1 and ETTh2 (hourly data) and ETTm1 (15-minute data). Each time step includes a target variable—transformer oil temperature—alongside six contextual power load features. The dataset is partitioned into training, validation, and testing sets in a 12/4/4-month configuration.
**Exchange**: This dataset consists of daily foreign exchange rate data spanning from 1990 to 2016, covering eight major currencies: Australian Dollar, British Pound, Canadian Dollar, Swiss Franc, Chinese Yuan, Japanese Yen, New Zealand Dollar, and Singapore Dollar. It serves as a valuable resource for economic and financial time-series modeling.
**ILI**: The Influenza-Like Illness (ILI) dataset, compiled by the U.S. Centers for Disease Control and Prevention (CDC), provides weekly reports from 2002 to 2021. It quantifies the prevalence of ILI cases as a ratio relative to total patient visits, thereby facilitating the study of epidemic trends and public health forecasting.
**Weather**: The weather dataset comprises high-frequency meteorological recordings taken at 10-minute intervals over the entirety of 2020. It includes 21 environmental variables, such as ambient temperature, relative humidity, and other atmospheric indicators, providing a comprehensive temporal profile for climate-related analyses.

Figure 7: Background Prompt 4.

To evaluate the effectiveness of the proposed Gaussian fuzzy mapping mechanism, we conduct a comprehensive analysis on four benchmark datasets (ETTh1, Exchange, ILI, and Weather) under four distinct background prompt settings. These prompt variants are generated by ChatGPT using the original background context as input. The results are reported in Table 8 and Table 9, respectively, where violin plots depict the distribution of prediction errors, measured by MSE and MAE, across different forecasting horizons and prompt configurations, with and without the Gaussian fuzzy mapping mechanism.

Table 8: Full results with Gaussian fuzzy mapping mechanism under four background prompts on four datasets.

| Methods | | Prompt 1 | | Prompt 2 | | Prompt 3 | | Prompt 4 | |
|---|---|---|---|---|---|---|---|---|---|
| Metric | | MSE | MAE | MSE | MAE | MSE | MAE | MSE | MAE |
| ETTh1 | 96 | **0.379** | **0.394** | 0.379 | 0.401 | 0.379 | 0.401 | 0.379 | 0.401 |
| | 192 | **0.430** | **0.424** | 0.436 | 0.432 | 0.436 | 0.432 | 0.436 | 0.432 |
| | 336 | **0.469** | **0.444** | 0.479 | 0.452 | 0.479 | 0.452 | 0.479 | 0.452 |
| | 720 | **0.479** | **0.467** | 0.497 | 0.483 | 0.497 | 0.483 | 0.497 | 0.483 |
| | AVG | **0.439** | **0.432** | 0.448 | 0.442 | 0.448 | 0.442 | 0.448 | 0.442 |
| Exchange | 96 | **0.083** | **0.203** | 0.086 | 0.205 | 0.086 | 0.205 | 0.086 | 0.205 |
| | 192 | **0.176** | **0.297** | 0.178 | 0.301 | 0.178 | 0.301 | 0.178 | 0.301 |
| | 336 | **0.319** | **0.409** | 0.324 | 0.412 | 0.324 | 0.412 | 0.324 | 0.412 |
| | 720 | **0.797** | **0.677** | 0.803 | 0.677 | 0.803 | 0.677 | 0.803 | 0.677 |
| | AVG | **0.344** | **0.397** | 0.348 | 0.399 | 0.348 | 0.399 | 0.348 | 0.399 |
| ILI | 24 | **2.446** | **0.926** | 2.631 | 1.055 | 2.632 | 1.056 | 2.654 | 1.054 |
| | 36 | **2.032** | **0.915** | 2.066 | 1.021 | 2.067 | 1.021 | 2.014 | 1.012 |
| | 48 | 2.054 | **0.904** | **2.020** | 1.035 | **2.020** | 1.035 | 2.051 | 1.023 |
| | 60 | **2.026** | **0.925** | 2.125 | 1.068 | 2.028 | 1.068 | 2.059 | 1.054 |
| | AVG | **2.140** | **0.918** | 2.211 | 1.045 | 2.187 | 1.045 | 2.195 | 1.036 |
| Weather | 96 | **0.159** | **0.202** | 0.168 | 0.210 | 0.169 | 0.211 | 0.169 | 0.211 |
| | 192 | **0.210** | **0.252** | 0.217 | 0.255 | 0.216 | 0.255 | 0.217 | 0.255 |
| | 336 | **0.270** | **0.296** | 0.276 | 0.300 | 0.275 | 0.300 | 0.276 | 0.298 |
| | 720 | **0.351** | **0.349** | 0.354 | 0.352 | 0.354 | 0.351 | 0.354 | 0.351 |
| | AVG | **0.248** | **0.275** | 0.254 | 0.279 | 0.254 | 0.279 | 0.254 | 0.279 |

Table 9: Full results without Gaussian fuzzy mapping mechanism under four background prompts on four datasets.

| Methods | | Prompt 1 | | Prompt 2 | | Prompt 3 | | Prompt 4 | |
|---|---|---|---|---|---|---|---|---|---|
| Metric | | MSE | MAE | MSE | MAE | MSE | MAE | MSE | MAE |
| ETT | 96 | 0.389 | 0.405 | **0.382** | **0.401** | **0.382** | **0.401** | 0.385 | 0.403 |
| | 192 | 0.443 | 0.436 | **0.434** | **0.431** | **0.434** | **0.431** | 0.439 | 0.435 |
| | 336 | 0.483 | 0.458 | **0.473** | 0.454 | 0.475 | **0.453** | 0.495 | 0.463 |
| | 720 | 0.516 | 0.494 | **0.482** | 0.482 | **0.482** | **0.476** | 0.486 | 0.480 |
| | AVG | 0.458 | 0.448 | **0.443** | 0.442 | **0.443** | **0.440** | 0.451 | 0.445 |
| Exchange | 96 | **0.084** | **0.205** | 0.088 | 0.207 | 0.088 | 0.207 | 0.089 | 0.208 |
| | 192 | 0.177 | 0.304 | 0.182 | 0.305 | 0.182 | 0.305 | **0.171** | **0.295** |
| | 336 | **0.321** | **0.411** | 0.371 | 0.442 | 0.368 | 0.440 | 0.344 | 0.427 |
| | 720 | **0.805** | **0.678** | 0.880 | 0.709 | 0.876 | 0.707 | 0.837 | 0.693 |
| | AVG | **0.347** | **0.399** | 0.380 | 0.416 | 0.379 | 0.415 | 0.360 | 0.406 |
| ILI | 24 | **2.615** | **1.057** | 2.801 | 1.099 | 2.798 | 1.098 | 2.799 | 1.098 |
| | 36 | 2.486 | 1.031 | 2.492 | 1.037 | **2.352** | **0.994** | 2.677 | 1.053 |
| | 48 | **2.505** | **1.034** | 2.856 | 1.077 | 2.930 | 1.046 | 2.658 | 1.069 |
| | 60 | **2.646** | **1.079** | 2.911 | 1.912 | 2.706 | 1.831 | 2.839 | 1.640 |
| | AVG | **2.563** | **1.050** | 2.765 | 1.281 | 2.697 | 1.242 | 2.743 | 1.215 |
| Weather | 96 | 0.166 | 0.210 | 0.170 | 0.213 | 0.170 | 0.213 | 0.170 | 0.214 |
| | 192 | **0.217** | 0.257 | 0.218 | **0.255** | 0.218 | 0.256 | 0.220 | 0.257 |
| | 336 | 0.275 | 0.300 | 0.275 | **0.299** | **0.274** | **0.299** | 0.279 | 0.300 |
| | 720 | **0.350** | 0.351 | 0.351 | **0.349** | 0.354 | **0.349** | 0.354 | **0.349** |
| | AVG | **0.252** | **0.279** | 0.254 | **0.279** | 0.254 | **0.279** | 0.256 | 0.280 |

As shown in Table 8 and Table 9, models augmented with the Gaussian fuzzy mapping mechanism consistently achieve lower MSE values compared to their non-fuzzy counterparts, particularly on complex and noisy datasets such as Exchange and ILI. This improvement is evident not only in terms of mean performance but also in the reduced variance of prediction errors, reflecting enhanced robustness and stability. For example, on the Exchange dataset, the incorporation of Gaussian fuzzy mapping mechanism significantly reduces both the central tendency and dispersion of the MSE distribution, demonstrating its effectiveness in handling high-frequency temporal fluctuations. A similar trend is observed in Table 8 and Table 9, where fuzzy-enhanced models exhibit lower MAE values across most datasets. The most notable improvement is found in the ILI dataset, where the

fuzzy component substantially lowers the prediction error, highlighting its advantage in modeling irregular and sparse medical time series. Although the performance gains are relatively modest on smoother datasets such as ETTh1 and Weather, the Gaussian fuzzy mapping mechanism still contributes to slight reductions in both mean error and variability.

Collectively, these results affirm that integrating the Gaussian fuzzy mapping enhances model generalization and resilience to error under varying background prompt scenarios. The findings underscore the importance of fuzzy mechanisms, particularly in scenarios involving uncertain or partially structured temporal patterns.

## F    SHOWCASES

To facilitate a clear comparison across different models, we present visual showcases of time series forecasting results on various datasets, including ETTh1 (Figure 8), ETTh2 (Figure 9), ETTm1 (Figure 10), ETTm2 (Figure 11), Exchange (Figure 12), ILI (Figure 13), and Weather (Figure 14).

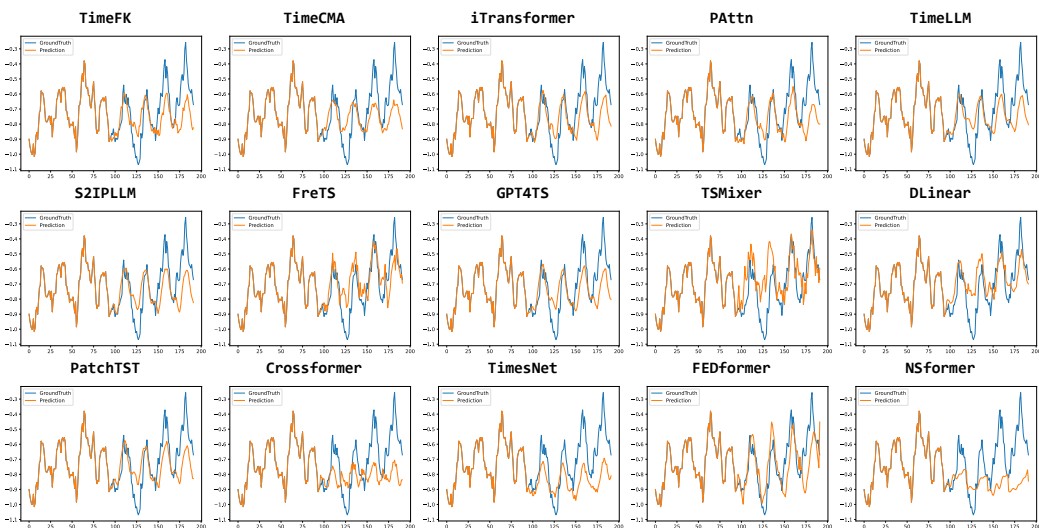

Figure 8: Visualization of ETTh1 predictions by different models under the input-96-predict-96 setting. The blue lines stand for the ground truth and the orange lines stand for predicted values.

## G    HYPERPARAMETER SENSITIVITY ON ENCODER DEPTH

### G.1    HYPERPARAMETER SENSITIVITY ANALYSIS ON TIME SERIES ENCODER DEPTH

To evaluate the impact of encoder depth on model performance, we conduct a hyperparameter sensitivity analysis by varying the depth of the time series encoder ($N_\mathcal{X} = 1$ to 4) across four benchmark datasets: ETTh1, Exchange, ILI, and Weather. As shown in Table 10, the model with encoder depth set to 2 consistently achieves the best or near-best performance across all datasets and forecasting horizons, as measured by MSE and MAE. Notably, on the ILI dataset, increasing the depth from 1 to 2 leads to a substantial reduction in average MSE (from 2.562 to 2.140) and MAE (from 1.045 to 0.918), demonstrating the importance of model capacity in handling complex seasonal patterns. Similarly, the Exchange dataset shows notable improvements with depth 2, especially for longer prediction lengths. In contrast, increasing the encoder depth beyond 2 generally does not yield further improvements and, in some cases, leads to performance degradation, possibly due to overfitting or optimization instability. These results suggest that an encoder depth of 2 offers a good balance between representation power and generalization, making it a robust default choice for TSF.

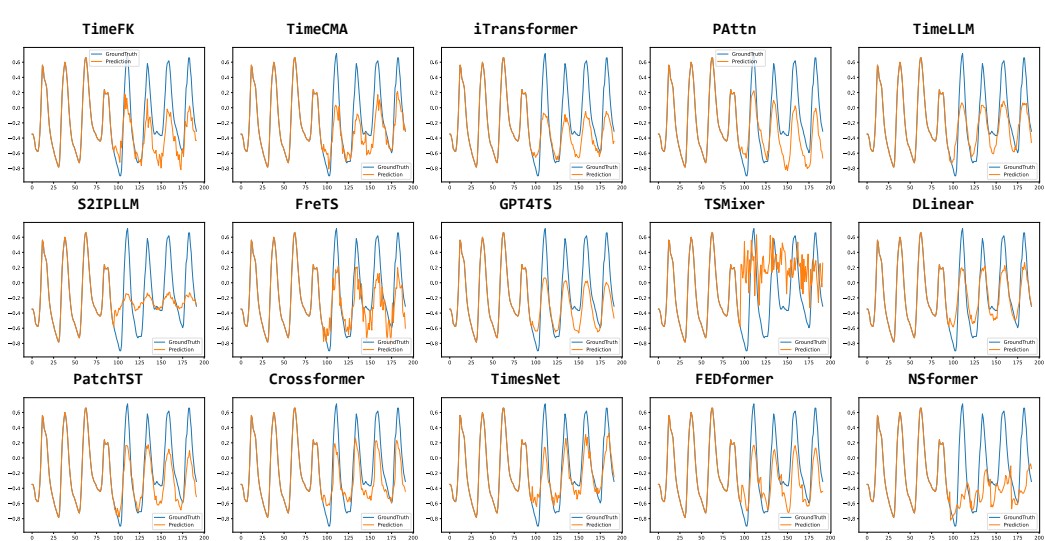

Figure 9: Visualization of ETTh2 predictions by different models under the input-96-predict-96 setting. The blue lines stand for the ground truth and the orange lines stand for predicted values.

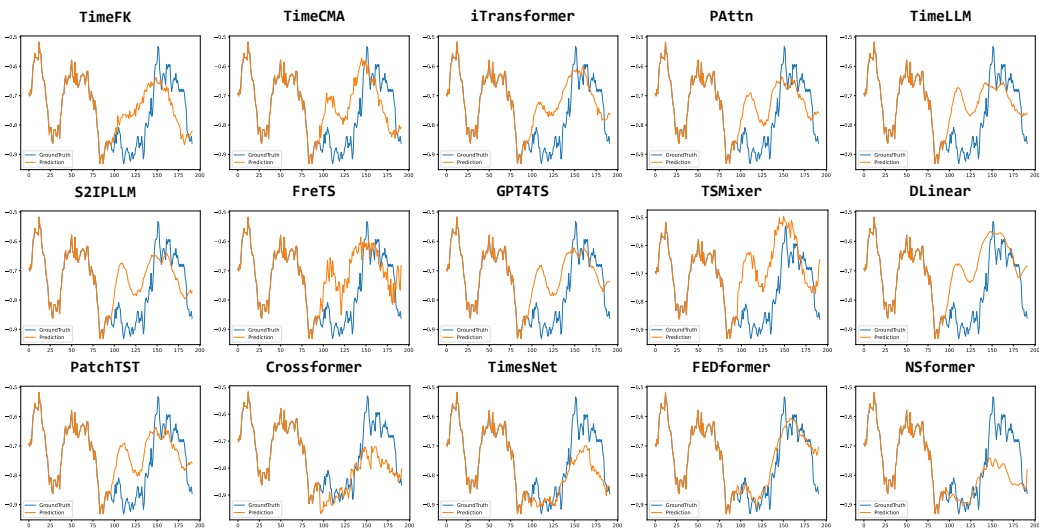

Figure 10: Visualization of ETTm1 predictions by different models under the input-96-predict-96 setting. The blue lines stand for the ground truth and the orange lines stand for predicted values.

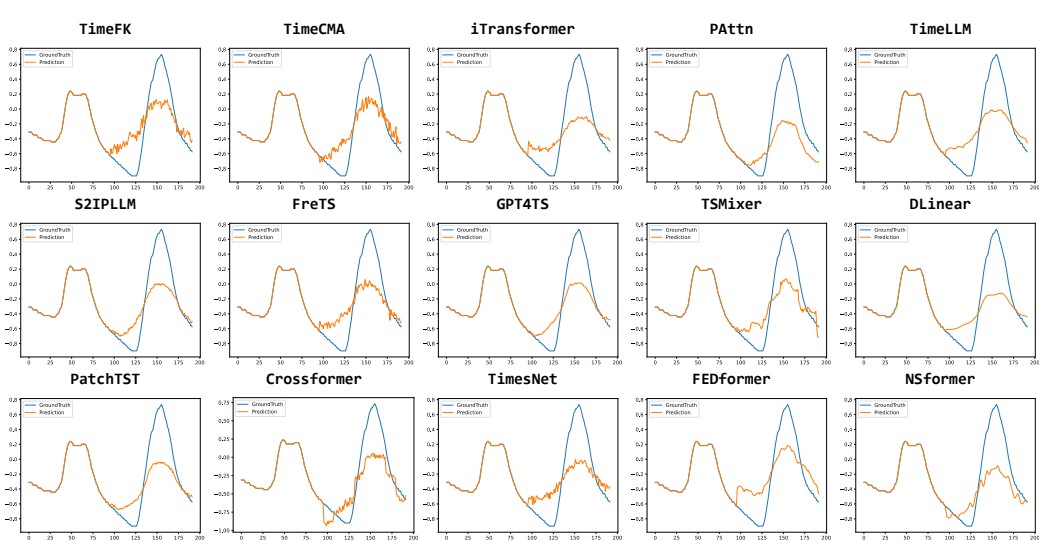

Figure 11: Visualization of ETTm2 predictions by different models under the input-96-predict-96 setting. The blue lines stand for the ground truth and the orange lines stand for predicted values.

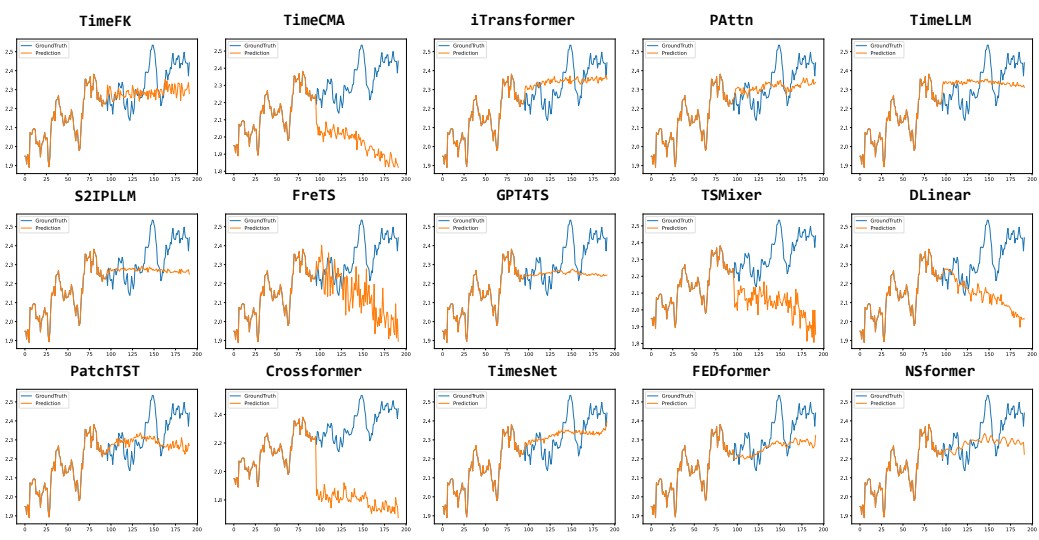

Figure 12: Visualization of Exchange predictions by different models under the input-96-predict-96 setting. The blue lines stand for the ground truth and the orange lines stand for predicted values.

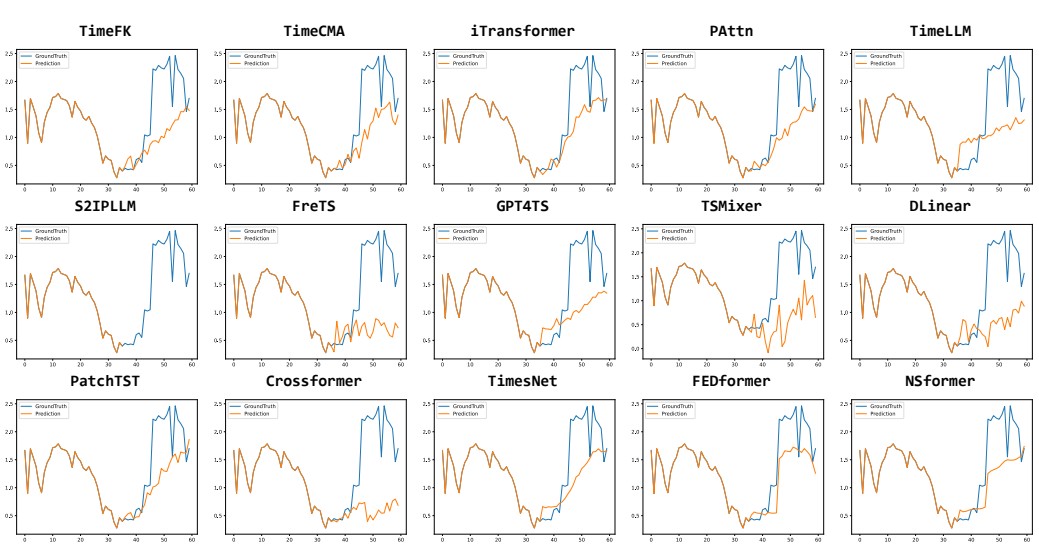

Figure 13: Visualization of ILI predictions by different models under the input-36-predict-24 setting. The blue lines stand for the ground truth and the orange lines stand for predicted values.

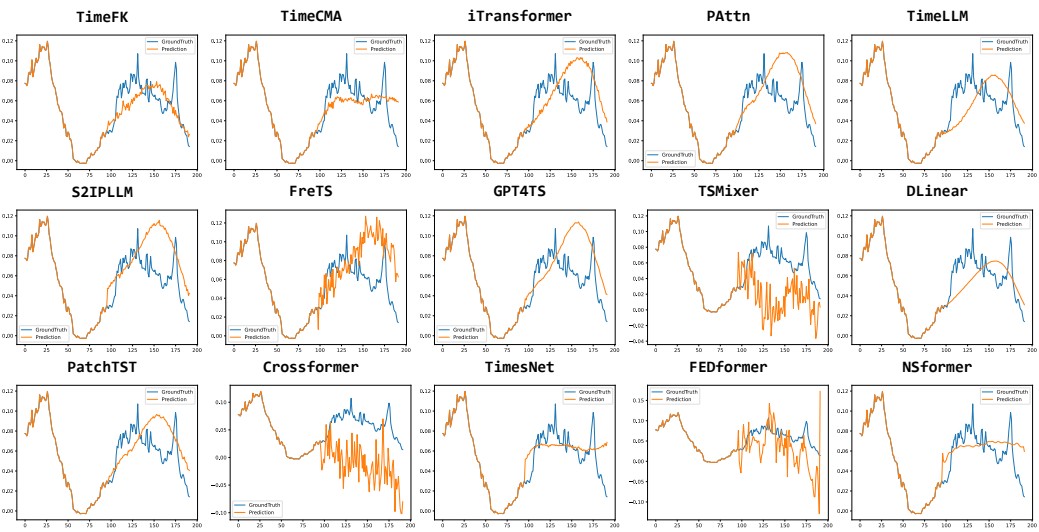

Figure 14: Visualization of Weather predictions by different models under the input-96-predict-96 setting. The blue lines stand for the ground truth and the orange lines stand for predicted values.

Table 10: Hyperparameter sensitivity on time series encoder depth.

| Methods | | $N_{\mathcal{X}}$ | | | | | | | |
|---|---|---|---|---|---|---|---|---|---|
| | | 1 | | 2 | | 3 | | 4 | |
| Metric | | MSE | MAE | MSE | MAE | MSE | MAE | MSE | MAE |
| ETTh1 | 96 | 0.382 | 0.401 | **0.379** | **0.394** | 0.384 | 0.403 | 0.385 | 0.403 |
| | 192 | 0.434 | 0.431 | **0.430** | **0.424** | 0.438 | 0.434 | 0.439 | 0.435 |
| | 336 | 0.476 | 0.453 | **0.469** | **0.444** | 0.489 | 0.460 | 0.495 | 0.463 |
| | 720 | 0.480 | 0.475 | **0.479** | **0.467** | 0.485 | 0.479 | 0.487 | 0.481 |
| | AVG | 0.443 | 0.440 | **0.439** | **0.432** | 0.449 | 0.444 | 0.452 | 0.446 |
| Exchange | 96 | 0.088 | 0.207 | **0.083** | **0.203** | 0.085 | 0.205 | 0.089 | 0.208 |
| | 192 | 0.186 | 0.308 | **0.176** | **0.297** | 0.178 | 0.302 | 0.171 | 0.295 |
| | 336 | 0.376 | 0.445 | **0.319** | **0.409** | 0.373 | 0.443 | 0.356 | 0.433 |
| | 720 | 0.875 | 0.707 | **0.797** | **0.677** | 0.867 | 0.706 | 0.838 | 0.693 |
| | AVG | 0.381 | 0.417 | **0.344** | **0.397** | 0.376 | 0.414 | 0.363 | 0.407 |
| ILI | 24 | 2.635 | 1.056 | **2.446** | **0.926** | 2.662 | 1.055 | 2.665 | 1.056 |
| | 36 | 2.468 | 1.021 | **2.032** | **0.915** | 2.424 | 1.012 | 2.417 | 1.013 |
| | 48 | 2.520 | 1.035 | **2.054** | **0.904** | 2.452 | 1.021 | 2.449 | 1.022 |
| | 60 | 2.627 | 1.068 | **2.026** | **0.925** | 2.565 | 1.054 | 2.564 | 1.056 |
| | AVG | 2.562 | 1.045 | **2.140** | **0.918** | 2.526 | 1.036 | 2.524 | 1.037 |
| Weather | 96 | 0.170 | 0.213 | **0.159** | **0.202** | 0.168 | 0.211 | 0.167 | 0.210 |
| | 192 | 0.221 | 0.257 | **0.210** | **0.252** | 0.220 | 0.257 | 0.215 | 0.255 |
| | 336 | 0.273 | 0.298 | **0.270** | **0.296** | 0.278 | 0.299 | 0.275 | 0.299 |
| | 720 | 0.352 | 0.349 | **0.351** | 0.349 | 0.354 | 0.349 | 0.352 | **0.348** |
| | AVG | 0.254 | 0.279 | **0.248** | **0.275** | 0.255 | 0.279 | 0.252 | 0.278 |

## G.2 HYPERPARAMETER SENSITIVITY ANALYSIS ON DUAL MODALITY ENCODER DEPTH

To evaluate the impact of the dual modality encoder depth on model performance, we conduct a sensitivity analysis by varying the number of encoder layers ($N_{\mathcal{P}}$) from 1 to 4. The results across four benchmark datasets (ETTh1, Exchange, ILI, and Weather) are summarized in Table 11, using MSE and MAE as evaluation metrics.

Overall, the model achieves the best or near-best performance when the encoder is configured with 2 layers. For the ETTh1 dataset, using two layers results in the lowest average MSE (0.439) and MAE (0.432), outperforming other settings across most forecasting horizons. A similar trend is observed on the Exchange dataset, where the two-layer configuration achieves significant improvements, particularly for long-term forecasting (e.g., 336 and 720), with an average MSE of 0.344 and MAE of 0.397. The ILI dataset exhibits the most pronounced gains: the two-layer model reduces the average MSE from 2.536 (1-layer) to 2.140 and MAE from 1.037 to 0.918. While the Weather dataset shows relatively stable performance across different configurations, the 2-layer setting still achieves the best overall average (MSE: 0.248, MAE: 0.275).

These results suggest that a dual modality encoder with two layers provides an optimal balance between model expressiveness and generalization ability, avoiding the underfitting risk of shallow architectures and the overfitting or increased complexity introduced by deeper ones. Hence, we adopt the two-layer encoder setting as the default configuration in our model.

## G.3 HYPERPARAMETER SENSITIVITY ANALYSIS ON BACKGROUND ENCODER DEPTH

To investigate the influence of background encoder depth ($N_{\mathcal{B}}$) on model performance, we vary the number of stacked blocks from 1 to 4 and evaluated the results across four standard datasets: ETTh1, Exchange, ILI, and Weather. As shown in Table 12, a depth of 2 blocks consistently yields superior or competitive results across all datasets and forecast horizons.

In the ETTh1 dataset, a 2-layer setting achieves the lowest average MSE (0.439) and MAE (0.432), outperforming both shallower and deeper configurations. The Exchange dataset shows a similar pattern, where the 2-layer configuration leads to the most pronounced gains, especially for long-term forecasting (e.g., 336 and 720 steps), with an average MSE of 0.344 and MAE of 0.397. For the ILI dataset, where temporal dynamics are highly nonlinear, the 2-layer model significantly reduces error metrics, improving average MSE from 2.536 to 2.140 and MAE from 1.037 to 0.918. The Weather

Table 11: Hyperparameter sensitivity on dual modality encoder depth.

| Methods | | $N_{\mathcal{P}}$ | | | | | | | |
| --- | --- | --- | --- | --- | --- | --- | --- | --- | --- |
| | | 1 | | 2 | | 3 | | 4 | |
| Metric | | MSE | MAE | MSE | MAE | MSE | MAE | MSE | MAE |
| ETTh1 | 96 | 0.382 | 0.402 | **0.379** | **0.394** | 0.383 | 0.402 | 0.383 | 0.402 |
| | 192 | 0.435 | 0.432 | **0.430** | **0.424** | 0.438 | 0.433 | 0.434 | 0.432 |
| | 336 | 0.487 | 0.458 | **0.469** | **0.444** | 0.486 | 0.458 | 0.487 | 0.458 |
| | 720 | 0.487 | 0.479 | **0.479** | **0.467** | 0.487 | 0.480 | 0.484 | 0.478 |
| | AVG | 0.448 | 0.443 | **0.439** | **0.432** | 0.449 | 0.443 | 0.447 | 0.443 |
| Exchange | 96 | 0.088 | 0.207 | **0.083** | **0.203** | 0.085 | 0.204 | 0.088 | 0.207 |
| | 192 | 0.181 | 0.304 | **0.176** | **0.297** | 0.180 | 0.304 | 0.179 | 0.303 |
| | 336 | 0.381 | 0.448 | **0.319** | **0.409** | 0.379 | 0.447 | 0.371 | 0.442 |
| | 720 | 0.866 | 0.704 | **0.797** | **0.677** | 0.868 | 0.704 | 0.870 | 0.706 |
| | AVG | 0.379 | 0.416 | **0.344** | **0.397** | 0.378 | 0.415 | 0.377 | 0.414 |
| ILI | 24 | 2.651 | 1.055 | **2.446** | **0.926** | 2.644 | 1.054 | 2.650 | 1.054 |
| | 36 | 2.434 | 1.013 | **2.032** | **0.915** | 2.434 | 1.013 | 2.432 | 1.013 |
| | 48 | 2.475 | 1.024 | **2.054** | **0.904** | 2.474 | 1.024 | 2.474 | 1.024 |
| | 60 | 2.586 | 1.057 | **2.026** | **0.925** | 2.587 | 1.058 | 2.589 | 1.058 |
| | AVG | 2.536 | 1.037 | **2.140** | **0.918** | 2.535 | 1.037 | 2.536 | 1.037 |
| Weather | 96 | 0.171 | 0.214 | **0.159** | **0.202** | 0.168 | 0.212 | 0.168 | 0.212 |
| | 192 | 0.220 | 0.257 | **0.210** | **0.252** | 0.217 | 0.255 | 0.217 | 0.255 |
| | 336 | 0.278 | 0.299 | **0.270** | **0.296** | 0.274 | 0.299 | 0.273 | 0.298 |
| | 720 | 0.353 | **0.348** | **0.351** | 0.349 | **0.351** | **0.348** | **0.351** | **0.348** |
| | AVG | 0.255 | 0.280 | **0.248** | **0.275** | 0.253 | 0.278 | 0.252 | 0.278 |

dataset exhibits relatively stable performance across all configurations; however, the 2-layer setting still obtains the lowest average MSE (0.248) and MAE (0.275).

These results suggest that increasing the depth beyond two layers does not consistently improve performance and may introduce unnecessary complexity. Conversely, using only a single layer may not capture sufficient modality interaction. Hence, we adopt a two-layer background encoder as the default configuration in our model due to its balanced effectiveness and efficiency across diverse tasks.

Table 12: Hyperparameter sensitivity on background encoder depth.

| Methods | | $N_{\mathcal{B}}$ | | | | | | | |
| --- | --- | --- | --- | --- | --- | --- | --- | --- | --- |
| | | 1 | | 2 | | 3 | | 4 | |
| Metric | | MSE | MAE | MSE | MAE | MSE | MAE | MSE | MAE |
| ETTh1 | 96 | 0.382 | 0.402 | **0.379** | **0.394** | 0.383 | 0.402 | 0.383 | 0.402 |
| | 192 | 0.435 | 0.432 | **0.430** | **0.424** | 0.438 | 0.433 | 0.434 | 0.432 |
| | 336 | 0.487 | 0.458 | **0.469** | **0.444** | 0.486 | 0.458 | 0.487 | 0.458 |
| | 720 | 0.486 | 0.479 | **0.479** | **0.467** | 0.488 | 0.480 | 0.485 | 0.478 |
| | AVG | 0.447 | 0.443 | **0.439** | **0.432** | 0.449 | 0.443 | 0.447 | 0.443 |
| Exchange | 96 | 0.088 | 0.207 | **0.083** | **0.203** | 0.085 | 0.204 | 0.088 | 0.207 |
| | 192 | 0.181 | 0.305 | **0.176** | **0.297** | 0.180 | 0.304 | 0.179 | 0.303 |
| | 336 | 0.380 | 0.448 | **0.319** | **0.409** | 0.379 | 0.447 | 0.371 | 0.442 |
| | 720 | 0.866 | 0.703 | **0.797** | **0.677** | 0.868 | 0.704 | 0.871 | 0.706 |
| | AVG | 0.379 | 0.416 | **0.344** | **0.397** | 0.378 | 0.415 | 0.377 | 0.415 |
| ILI | 24 | 2.652 | 1.055 | **2.446** | **0.926** | 2.643 | 1.054 | 2.651 | 1.055 |
| | 36 | 2.433 | 1.013 | **2.032** | **0.915** | 2.434 | 1.013 | 2.432 | 1.013 |
| | 48 | 2.475 | 1.024 | **2.054** | **0.904** | 2.474 | 1.024 | 2.474 | 1.024 |
| | 60 | 2.586 | 1.057 | **2.026** | **0.925** | 2.587 | 1.058 | 2.589 | 1.058 |
| | AVG | 2.536 | 1.037 | **2.140** | **0.918** | 2.535 | 1.037 | 2.537 | 1.037 |
| Weather | 96 | 0.171 | 0.214 | **0.159** | **0.202** | 0.168 | 0.212 | 0.172 | 0.215 |
| | 192 | 0.220 | 0.256 | **0.210** | **0.252** | 0.217 | 0.255 | 0.217 | 0.255 |
| | 336 | 0.278 | 0.299 | **0.270** | **0.296** | 0.274 | 0.299 | 0.273 | 0.298 |
| | 720 | 0.352 | 0.349 | 0.351 | 0.349 | 0.351 | **0.347** | **0.350** | 0.348 |
| | AVG | 0.255 | 0.280 | **0.248** | **0.275** | 0.252 | 0.278 | 0.253 | 0.279 |

## H HYPERPARAMETER SENSITIVITY ANALYSIS ON DECODER DEPTH

Table 13 presents the results of a hyperparameter sensitivity analysis on decoder depth across four benchmark datasets (ETTh1, Exchange, ILI, and Weather) and multiple forecast lengths. Each cell reports the MSE and MAE for decoder depths ranging from 1 to 4. The bolded values indicate the best performance in each row. The results show that a decoder depth of 1 often yields the best or competitive results, especially on the ILI and ETTh1 datasets. For example, in the ILI dataset, depth 1 achieves the lowest average MSE (2.140) and MAE (0.918), outperforming deeper configurations. In contrast, the Exchange dataset benefits slightly from deeper decoders at longer prediction horizons, while the Weather dataset demonstrates relatively stable performance across depths. These findings suggest that shallow decoders are often sufficient and may generalize better, while deeper decoders could be dataset-dependent and require careful tuning.

Table 13: Hyperparameter sensitivity on decoder depth.

| Methods | | Depth | | | | | | | |
|---|---|---|---|---|---|---|---|---|---|
| | | 1 | | 2 | | 3 | | 4 | |
| Metric | | MSE | MAE | MSE | MAE | MSE | MAE | MSE | MAE |
| ETTh1 | 96 | **0.379** | **0.394** | 0.383 | 0.405 | 0.384 | 0.406 | 0.384 | 0.405 |
| | 192 | **0.430** | **0.424** | 0.438 | 0.434 | 0.442 | 0.436 | 0.444 | 0.437 |
| | 336 | **0.469** | **0.444** | 0.489 | 0.458 | 0.487 | 0.458 | 0.479 | 0.457 |
| | 720 | **0.479** | **0.467** | 0.496 | 0.483 | 0.493 | 0.481 | 0.494 | 0.482 |
| | AVG | **0.439** | **0.432** | 0.451 | 0.445 | 0.452 | 0.445 | 0.450 | 0.445 |
| Exchange | 96 | **0.083** | **0.203** | 0.084 | 0.204 | 0.086 | 0.205 | 0.085 | 0.206 |
| | 192 | **0.176** | **0.297** | 0.182 | 0.304 | 0.178 | 0.301 | 0.179 | 0.303 |
| | 336 | 0.319 | 0.409 | 0.322 | 0.411 | 0.307 | **0.401** | **0.307** | **0.401** |
| | 720 | **0.797** | **0.677** | 0.807 | 0.680 | 0.815 | 0.686 | 0.802 | 0.679 |
| | AVG | 0.344 | **0.397** | 0.349 | 0.400 | 0.346 | 0.399 | **0.343** | **0.397** |
| ILI | 24 | **2.446** | **0.926** | 2.678 | 1.055 | 2.732 | 1.064 | 2.751 | 1.050 |
| | 36 | **2.032** | **0.915** | 2.329 | 0.994 | 2.359 | 1.003 | 2.396 | 1.013 |
| | 48 | **2.054** | **0.904** | 2.263 | 0.983 | 2.243 | 0.982 | 2.262 | 0.989 |
| | 60 | **2.026** | **0.925** | 2.364 | 1.008 | 2.369 | 1.016 | 2.381 | 1.021 |
| | AVG | **2.140** | **0.918** | 2.408 | 1.010 | 2.426 | 1.016 | 2.448 | 1.018 |
| Weather | 96 | **0.159** | **0.202** | 0.181 | 0.222 | 0.169 | 0.212 | 0.170 | 0.213 |
| | 192 | **0.210** | **0.252** | 0.217 | 0.256 | 0.217 | 0.256 | 0.219 | 0.259 |
| | 336 | **0.270** | **0.296** | 0.277 | 0.302 | 0.276 | 0.301 | 0.278 | 0.304 |
| | 720 | **0.351** | **0.349** | 0.357 | 0.354 | 0.353 | 0.351 | 0.353 | 0.354 |
| | AVG | **0.248** | **0.275** | 0.258 | 0.283 | 0.254 | 0.280 | 0.255 | 0.282 |

## I HYPERPARAMETER SENSITIVITY ANALYSIS ON LLM DEPTH

Table 14 presents the performance of LLMs with different depths across four datasets (ETTh1, Exchange, ILI, and Weather), evaluated using MSE and MAE. As the LLM depth increases, we observe a general trend of performance improvement, particularly when increasing from 0 to 4 layers. Specifically, the 4-layer LLM achieves the lowest or near-lowest MSE and MAE in most cases. For instance, on the ETTh1 dataset, the 4-layer LLM achieves the best average MSE (0.439) and MAE (0.432) across all prediction lengths. Similar performance gains are also observed on the Exchange and ILI datasets.

While the 8-layer LLM occasionally shows marginal improvements (e.g., on the 96-length prediction of the Exchange dataset), the performance gains are limited and come at the cost of significantly increased model complexity and computation. Therefore, to strike a balance between model performance and complexity, we choose the 4-layer LLM as the default configuration. It offers substantial improvements over shallower models while avoiding the overhead and potential overfitting risks of deeper networks.

Moreover, the 4-layer LLM demonstrates stable performance across all datasets, indicating better generalization capability.

Table 14: Hyperparameter sensitivity on LLM depth.

| Methods | | Depth | | | | | | | |
|---|---|---|---|---|---|---|---|---|---|
| | | **0** | | **2** | | **4** | | **8** | |
| Metric | | MSE | MAE | MSE | MAE | MSE | MAE | MSE | MAE |
| ETTh1 | 96 | 0.385 | 0.402 | 0.387 | 0.403 | **0.379** | **0.394** | 0.380 | 0.402 |
| | 192 | 0.434 | 0.431 | 0.435 | 0.432 | **0.430** | **0.424** | 0.437 | 0.432 |
| | 336 | 0.477 | 0.453 | 0.474 | 0.453 | **0.469** | **0.444** | 0.480 | 0.452 |
| | 720 | 0.514 | 0.489 | 0.478 | 0.474 | **0.479** | **0.467** | 0.495 | 0.481 |
| | AVG | 0.452 | 0.444 | 0.444 | 0.441 | **0.439** | **0.432** | 0.448 | 0.442 |
| Exchange | 96 | 0.086 | 0.205 | 0.088 | 0.207 | **0.083** | **0.203** | **0.083** | **0.203** |
| | 192 | 0.177 | 0.300 | 0.189 | 0.312 | **0.176** | **0.297** | 0.178 | 0.301 |
| | 336 | 0.332 | 0.417 | 0.351 | 0.426 | **0.319** | **0.409** | 0.326 | 0.413 |
| | 720 | 0.848 | 0.696 | 0.832 | 0.688 | **0.797** | 0.677 | 0.799 | **0.676** |
| | AVG | 0.361 | 0.404 | 0.365 | 0.408 | **0.344** | **0.397** | 0.347 | 0.398 |
| ILI | 24 | 2.777 | 1.094 | 2.743 | 1.092 | **2.446** | **0.926** | 2.807 | 1.100 |
| | 36 | 2.471 | 1.022 | 2.464 | 1.019 | **2.032** | **0.915** | 2.354 | 0.995 |
| | 48 | 2.534 | 1.039 | 2.528 | 1.039 | **2.054** | **0.904** | 2.319 | 0.987 |
| | 60 | 2.618 | 1.067 | 2.690 | 1.082 | **2.026** | **0.925** | 2.503 | 1.037 |
| | AVG | 2.600 | 1.056 | 2.606 | 1.058 | **2.140** | **0.918** | 2.496 | 1.030 |
| Weather | 96 | 0.171 | 0.214 | 0.170 | 0.212 | **0.159** | **0.202** | 0.168 | 0.210 |
| | 192 | 0.218 | 0.256 | 0.219 | 0.256 | **0.210** | **0.252** | 0.221 | 0.258 |
| | 336 | 0.273 | 0.298 | 0.274 | 0.299 | **0.270** | **0.296** | 0.274 | 0.299 |
| | 720 | **0.350** | **0.349** | 0.351 | 0.349 | 0.351 | 0.349 | 0.353 | 0.351 |
| | AVG | 0.253 | 0.279 | 0.253 | 0.279 | **0.248** | **0.275** | 0.254 | 0.280 |

# J    HYPERPARAMETER SETTINGS

In TimeFK, we set the number of layers for the LLM to 4, and unified the number of layers for all encoders to 2 in order to sufficiently encode both linguistic and numerical information. Additionally, we configured the decoder with a single layer to perform fuzzy-aware attention decoder. The learning rate was set to 1e-4, following the setting in TimesNet (Wu et al., 2023).

# K    COMPUTATIONAL COST

To quantify the concern regarding computational overhead introduced by the tri-branch encoding and fuzzy mapping, we conducted additional experiments on the Weather dataset, which includes 21 variables. In particular, we measured memory usage during inference with a prediction length of 720, using GPT-2 as the underlying LLM. The results are presented in the table below:

Table 15: Memory usage analysis on the Weather dataset (prediction length 720).

| Method | TimeFK | TimeCMA | iTransformer | PAttn | TimeLLM | S2IPLLM | FreTS | GPT4TS | TSMixer | DLinear | PatchTST | Crossformer | TimesNet | FEDformer | NSformer |
|---|---|---|---|---|---|---|---|---|---|---|---|---|---|---|---|
| Memoty Usage (MB) | 432.13 | 566.04 | 10.84 | 13.48 | 677.82 | 508.93 | 12.96 | 266.85 | 0.73 | 0.53 | 18.49 | 61.85 | 2288.57 | 22.37 | 16.57 |

These results quantitatively illustrate the memory footprint associated with our proposed design. As shown in Table 15, our method (TimeFK) introduces moderate memory usage (432.13 MB) during inference with a prediction length of 720. Although this is higher than some lightweight baselines such as DLinear (0.53 MB) and iTransformer (10.84 MB), it remains significantly lower than other LLM-based or attention-heavy models, such as TimeLLM (677.82 MB) and TimesNet (2288.57 MB). Notably, compared to other strong LLM baselines like S2IPLLM (508.93 MB), our method achieves enhanced modeling capability with acceptable overhead, especially considering the added expressiveness brought by tri-branch encoding and fuzzy mapping.

Overall, while the use of GPT-2 and the proposed modules do introduce some additional computational cost, we believe the trade-off is justified by the significant performance gains offered by our architecture.

## L  THE GENERALITY OF TIMEFK

Our framework extracts hidden representations from intermediate layers of LLMs rather than relying solely on their final outputs. This prevents us from experimenting with closed-source models such as GPT-4, which do not expose internal layer activations. To further examine generality beyond GPT-2, we replaced it with two open-source LLM backbones, LlaMA-3.2-1B and Qwen3-0.6B. The detailed results are reported in the table below.

| Methods | | GPT-2 | | LlaMA-3.2-1B | | Qwen3-0.6B | |
|---|---|---|---|---|---|---|---|
| Metric | | MSE | MAE | MSE | MAE | MSE | MAE |
| ETTh1 | 96 | **0.379** | **0.394** | 0.380 | 0.401 | **0.379** | 0.399 |
| | 192 | **0.430** | **0.424** | 0.438 | 0.435 | 0.438 | 0.431 |
| | 336 | **0.469** | **0.444** | 0.475 | 0.451 | 0.476 | 0.452 |
| | 720 | 0.479 | **0.467** | **0.470** | 0.470 | 0.526 | 0.496 |
| | AVG | **0.439** | **0.432** | 0.441 | 0.439 | 0.454 | 0.445 |
| Exchange | 96 | **0.083** | **0.203** | 0.085 | 0.204 | **0.083** | **0.203** |
| | 192 | **0.176** | **0.297** | 0.191 | 0.309 | 0.180 | 0.302 |
| | 336 | **0.319** | **0.409** | 0.350 | 0.428 | 0.337 | 0.418 |
| | 720 | **0.797** | **0.677** | 0.859 | 0.701 | 0.885 | 0.710 |
| | AVG | **0.344** | **0.397** | 0.371 | 0.410 | 0.371 | 0.408 |
| ILI | 24 | **2.446** | **0.926** | 2.627 | 1.041 | 2.688 | 1.051 |
| | 36 | **2.032** | **0.915** | 2.338 | 0.991 | 2.341 | 0.990 |
| | 48 | **2.054** | **0.904** | 2.295 | 0.984 | 2.405 | 1.008 |
| | 60 | **2.026** | **0.925** | 2.461 | 1.032 | 2.437 | 1.023 |
| | AVG | **2.140** | **0.918** | 2.430 | 1.012 | 2.468 | 1.018 |
| Weather | 96 | **0.159** | **0.202** | 0.180 | 0.223 | 0.182 | 0.225 |
| | 192 | **0.210** | **0.252** | 0.221 | 0.259 | 0.220 | 0.257 |
| | 336 | **0.270** | **0.296** | 0.275 | 0.300 | 0.280 | 0.310 |
| | 720 | **0.351** | **0.349** | 0.355 | 0.348 | 0.360 | 0.350 |
| | AVG | **0.248** | **0.275** | 0.258 | 0.283 | 0.261 | 0.286 |

These results demonstrate that our framework maintains competitive and consistent performance across different backbone LLMs, confirming its general applicability beyond GPT-2.

## M  CODE

Code is available at `https://anonymous.4open.science/r/TimeFK-B2DF`.

