# OpenReview forum: "TimeFK: Towards Time Series Forecasting via Treating LLMs as Fuzzy Key"
_ICLR.cc/2026/Conference — ICLR 2026 Conference Withdrawn Submission_

### Official Review · Reviewer_4HqS · 2025-10-24

**Soundness:** 2
**Presentation:** 2
**Contribution:** 1
**Rating:** 2
**Confidence:** 5

**Summary:**

This paper works on multivariate time series forecasting (TSF), aiming to utilize the well-learned knowledge of LLMs to improve the performance of TSF in a multi-modal manner. Specifically, this paper presents TimeFK, a TSF framework that leverages LLM as "fuzzy keys" to enhance the prediction capabilities. The main technical components comprise: (1) a tri-branch multi-modal encoder (time series, statistical, and background encodings), (2) a Gaussian fuzzy mapping mechanism to reduce prompt noise, and (3) a fuzzy-aware attention decoder to avoid representation entanglement. Extensive experiments have been conducted and the reported performance is good.

**Strengths:**

1. Extensive experiments.
2. The reported performance is good.

**Weaknesses:**

1. This paper appears to be an incremental method built upon TimeCMA, it's necessary to provide a detailed comparisons between TimeFK and TimeCMA, particularly their differences.
2. The motivation of introducing Background Encoding Branch is unclear, as well as its role played in Cross-Modality Fusing module.
3. In L053, authors claim TimeCMA-like method has an efficiency bottleneck, however, (1) TimeCMA only has 2 branches, while TimeFK has 3 branches; (2) there is no efficiency comparisons (but not memory comparisons provided in appendix) to demonstrate that TimeFK is efficient; (3) why not put Background Input in front of Statistical Input to be a single prompt, the former part is short and fixed, and we can also use KV Cache to avoid replicated calculations.
4. The main motivation of treating LLMs as "fuzzy keys" is underdeveloped and lacks rigorous theoretical grounding, (1) the paper fails to clearly define what constitutes a "fuzzy key" in the context of TSF, i.e., how LLMs inherently align with fuzzy set theory remains unclear; (2) there are no clues to validate the claim that this component can not only reduce noise but also perform better cross-modality fusion that previous methods (like TimeCMA).
5. The Fuzzy-aware Attention Decoder looks like a standard Transformer Decoder Block, without anyother design. In general, the novelty of this paper is quite limited.
6. Regarding baselines, (1) TimeCMA is not the current state-of-the-arts, there are some stronger models like TimeXer; (2) in quantitative results, the reported values of TimeCMA is much worse than those reported in their paper, i.e., the comparison is unfair and invalid.

**Questions:**

N.A.

---

### Official Review · Reviewer_LV4R · 2025-10-28

**Soundness:** 2
**Presentation:** 2
**Contribution:** 1
**Rating:** 2
**Confidence:** 5

**Summary:**

This paper studies an important problem of time series forecasting based on large language models (LLMs). The authors design a tri-branch multi-modal encoding scheme with numerical and linguistic representations. In addition, a Gaussian fuzzy mapping mechanism is proposed to alleviate the noise of LLMs. Experiments show the effectiveness of the proposed method to some extent.

**Strengths:**

1. The paper is well-written and easy to follow.
2. A tri-branch multi-modal encoder is proposed to learn comprehensive representations of time series.
3. Experiments are conducted to show the effectiveness of the proposed method.

**Weaknesses:**

1. Although the baseline comparison shows that TimeFK has better performance than existing methods. However, it is strange that the prediction of TimeFK does not follow the ground truth in Figure 8. Especially, DLinear, FreTS, and FEDformer show that their prediction can more accurately trace the ground truth. This raises a concern about whether TimeFK really works for time series forecasting.
2. More recent baselines are required as follows.

[1]. UniTime: A Language-Empowered Unified Model for Cross-Domain Time Series Forecasting, WWW 2024.

[2]. One fits all: Power general time series analysis by pretrained LM, NeurIPS 2023.

[3]. Sundial: A Family of Highly Capable Time Series Foundation Models, ICML 2025.

[4]. Timer-XL: Long-Context Transformers for Unified Time Series Forecasting, ICLR 2025.

[5]. DUET: Dual Clustering Enhanced Multivariate Time Series Forecasting, SIGKDD 2025.

3. After carefully checking the baselines, the proposed TimeFK seems to be an incremental version of TimeCMA with an additional background encoding branch. Further, the prompt templates shown in Figure 3 are almost the same as those in TimeCMA. This implies insufficient technical improvement over existing papers.

4. It would be better to include a case study to intuitively show the effectiveness of the proposed Gaussian fuzzy mapping mechanism as this is the major argument.

5. The authors argue that existing LLM-based methods may have low inference speed. However, there are no experiments regarding efficiency (e.g., inference time and FLOPs) in the major component of the paper. Only the memory comparison in Table 15 is not enough to show the efficiency of the proposed method. It is also suggested to include the training time comparison. The efficiency experiments are suggested to be conducted on multiple datasets with different prediction lengths.

**Questions:**

Please see the weaknesses.

---

### Official Review · Reviewer_15sJ · 2025-10-30

**Soundness:** 3
**Presentation:** 3
**Contribution:** 3
**Rating:** 10
**Confidence:** 3

**Summary:**

To bridge the gap between LLM-based methods and deep learning approaches due to their inherent differences, this work propose TimeFK, an innovative TSF framework that uses LLMs as “fuzzy keys” to activate forecasting capabilities. A wealth of empirical evidence demonstrates the effectiveness of the method proposed in this work.

**Strengths:**

1. The research motivation is relatively clear. The understanding of the current existing research is profound.

2. The proposed architecture is very interesting and can bring new insights to the field.

3. The discussion of the experiment is sufficient and reasonable

**Weaknesses:**

1. Can other LLMs except GPT2 be selected as the control for the experiment?

2. Whether the viewpoints presented in this work are transferable, such as being expanded in the prediction of spatio-temporal data?

3. The discussion on training efficiency and prediction efficiency should be analyzed.

4. Sec 2 should be organized more logically to demonstrate the differences between this paper and the existing works.

**Questions:**

See Weaknesses

---

### Official Review · Reviewer_c9WE · 2025-11-01

**Soundness:** 1
**Presentation:** 1
**Contribution:** 1
**Rating:** 0
**Confidence:** 4

**Summary:**

The authors proposed a new multimodal method for time series forecasting and evaluated it on 4 ETTs, Exchange, ILI, and Weather  benchmarks.

**Strengths:**

- The proposed pipeline is easy to understand.

**Weaknesses:**

### **Baselines and Related Work**
For the multimodal time series forecasting method, I suggest the author add a discussion and comparison with [1-4].


[1] Time-VLM: Exploring Multimodal Vision-Language Models for Augmented Time Series Forecasting

[3] Multi-Modal View Enhanced Large Vision Models for Long-Term Time Series Forecasting

[3] Teaching Time Series to See and Speak: Forecasting with Aligned Visual and Textual Perspectives


### **Dataset**
The author only evaluated their method on ETT, Exchange, ILI, and Weather.  There is no short-term forecasting dataset, e.g., M4 and PEMS, and other commonly used datasets need to be evaluated.


### **Wriring**

There are lots of paragraphs with GPT-style words.

**Questions:**

Please refer to Weakness.

**Details Of Ethics Concerns:**

Some sections are heavily GPT-style words. Additionally, I attempted to use a third-party tool to verify it, which indicated that it was 100% AI-generated writing. Although the tool is not fully conclusive, it serves as a reference.

For example,

> 2 RELATED WORK
> FEDformer (Zhou et al., 2022) incorporated a mixture-of-experts design to enhance trend and seasonal decomposition and
proposed a sparse attention mechanism in the frequency domain to balance efficiency and accuracy.
TimesNet (Wu et al., 2023) decomposed time series into periodic segments and modeled intra- and inter-period interactions by Inception blocks, facilitating generalized time series modeling. Crossformer (Zhang & Yan, 2023) captured dependencies across both temporal and variable dimensions using an attention mechanism. PatchTST (Nie et al., 2023) patched time series into subsequences, preserving local semantics and enabling long-range temporal modeling. DLinear (Zeng et al., 2023) introduced a simple one-layer linear model that achieved high accuracy through direct multistep prediction. TSMixer (Chen et al., 2023) extracted temporal and feature-wise information by stacking multilayer perceptrons (MLPs) across mixed time-feature dimensions. To address the limitations of pointwise mapping and the information bottleneck of MLP-based approaches, FreTS (Yi et al., 2023) applied MLPs in the frequency domain to improve global dependency modeling. iTransformer (Liu et al., 2024b) leveraged dimension inversion to enhance long-sequence handling, mitigating performance degradation and reducing computational overhead.

> Tab. 1
I also read Tab.1, which is very similar to GPT-style.

---

### Note · Authors · 2025-11-17

I have read and agree with the venue's withdrawal policy on behalf of myself and my co-authors.